# Pigment cell progenitor heterogeneity and reiteration of developmental signaling underlie melanocyte regeneration in zebrafish

William Tyler Frantz[1,2], Sharanya Iyengar[1,2], James Neiswender[1,2], Alyssa Cousineau[1], René Maehr[1], Craig J Ceol[1,2]*

[1]Program in Molecular Medicine, University of Massachusetts Medical School, Worcester, United States; [2]Department of Molecular, Cell and Cancer Biology, University of Massachusetts Medical School, Worcester, United States

*For correspondence:
craig.ceol@umassmed.edu

Competing interest: The authors declare that no competing interests exist.

**Abstract** Tissue-resident stem and progenitor cells are present in many adult organs, where they are important for organ homeostasis and repair in response to injury. However, the signals that activate these cells and the mechanisms governing how these cells renew or differentiate are highly context-dependent and incompletely understood, particularly in non-hematopoietic tissues. In the skin, melanocyte stem and progenitor cells are responsible for replenishing mature pigmented melanocytes. In mammals, these cells reside in the hair follicle bulge and bulb niches where they are activated during homeostatic hair follicle turnover and following melanocyte destruction, as occurs in vitiligo and other skin hypopigmentation disorders. Recently, we identified melanocyte progenitors in adult zebrafish skin. To elucidate mechanisms governing melanocyte progenitor renewal and differentiation we analyzed individual transcriptomes from thousands of melanocyte lineage cells during the regeneration process. We identified transcriptional signatures for progenitors, deciphered transcriptional changes and intermediate cell states during regeneration, and analyzed cell–cell signaling changes to discover mechanisms governing melanocyte regeneration. We identified KIT signaling via the RAS/MAPK pathway as a regulator of melanocyte progenitor direct differentiation and asymmetric division. Our findings show how activation of different subpopulations of *mitfa*-positive cells underlies cellular transitions required to properly reconstitute the melanocyte pigmentary system following injury.

## Editor's evaluation

This valuable study advances our understanding of heterogeneous transcriptomic states and genetic requirements of skin-resident pigment cells and pigment cell progenitors in adult zebrafish, relevant to regenerative biology and melanoma origins. The single-cell and bioinformatic analyses and the use of mutants and regeneration assays are carefully done and appropriately interpreted. The work provides useful new observations that will be of interest to researchers focused on the basic biology of adult pigmentary phenotypes and their homeostasis, as well as those pursuing translational aspects of regeneration and melanoma origins and treatments.

## Introduction

Adult stem and progenitor cells are responsible for adult tissue maintenance and critical in recovery from injury. These cells are maintained through renewal, often self-renewal, and respond to tissue

injury by proliferating or differentiating. Melanocyte stem cells (McSCs), which are found in the hair follicle niche in mammals, are adult stem cells that replenish melanocytes, which ultimately impart pigment to hair and skin (*Nishimura et al., 2002*; *Slominski et al., 2005*). Mechanisms governing adult stem and progenitor cell behavior are critical to maintaining proper tissue function as well as recovery from injury or disease.

Zebrafish, with their characteristic stripes of pigment-retaining dermal melanocytes, have emerged as a useful model to study melanocyte regeneration. Ablation of embryonic and adult melanocytes results in generation of new melanocytes from tissue-resident stem or progenitor cells (*Hultman et al., 2009*; *O'Reilly-Pol and Johnson, 2013*; *Rawls and Johnson, 2000*; *Yang et al., 2007*). Cells involved in melanocyte regeneration were identified in adult zebrafish stripes as unpigmented cells that are admixed with mature melanocytes and express *mitfa*, the zebrafish ortholog of the MITF melanocyte lineage regulator (*Iyengar et al., 2015*). These zebrafish melanocyte progenitors respond to injury by either differentiating into mature melanocytes to reconstitute the skin's pigment pattern or dividing asymmetrically to generate an unpigmented daughter and another daughter that ultimately differentiates into a melanocyte. Additional cells divide symmetrically, and they are hypothesized to replenish the progenitor pool of cells for subsequent responses to injury. Notably, unpigmented daughter cells from progenitor divisions can, following subsequent injury, directly differentiate, indicating a plasticity of fates that can be adopted by progenitors (*Iyengar et al., 2015*). Other tissue-resident cells, such as murine basal epidermal stem cells, can also adopt different fates during regeneration. These epidermal stem cells appear initially uncommitted, with environmental signals likely to trigger self-renewal or differentiation (*Rompolas et al., 2016*).

Identification of zebrafish melanocyte progenitors facilitated our investigation of mechanisms governing differentiation. In vivo imaging revealed that the unpigmented progenitors upregulate WNT signaling prior to undergoing differentiation, and animals treated with WNT inhibitor fail to regenerate. Meanwhile, symmetric and asymmetric divisions were unaffected, demonstrating a fate-specific requirement for WNT signaling (*Iyengar et al., 2015*). Like in the zebrafish, previous murine studies have identified coordinated WNT signaling as a key regulator of differentiation during melanocyte regeneration (*Rabbani et al., 2011*). More recently, profiling of stem cells and melanocyte single-cell transcriptomes from regenerating hair follicles has revealed coordinated activation of WNT/BMP signaling as a gate governing differentiation (*Infarinato et al., 2020*). These studies underline the conserved mechanisms regulating melanocyte regeneration in murine and zebrafish models, emphasizing the utility of studying adult progenitors in zebrafish.

While the mechanisms governing stem cell fates during regeneration have been elucidated in some tissues, adult stem cell identities and responses to injury are incompletely understood. Some tissue-resident stem cells, such as epidermal hair follicle and basal epidermal stem cells, rely on positional cues for fate determination, suggestive of a stochastic model (*Rompolas and Greco, 2014*; *Rompolas et al., 2016*). Yet, other tissue-resident stem cells, such as hematopoietic and lung epithelial stem cells, appear to be organized in a hierarchical manner with a self-renewing common multipotent stem cell giving rise to lineage-committed progeny (*Laurenti and Göttgens, 2018*; *McQualter et al., 2010*). Even within the same stem cell pool the behaviors can be context dependent. In response to injury, murine McSCs will prioritize differentiation over self-renewal, but in normal hair follicle turnover these cells engage in balanced differentiation and self-renewal (*Chou et al., 2013*; *Nishimura et al., 2002*). It is currently unknown if the different McSC and melanocyte progenitor behaviors observed in murine and zebrafish regeneration studies are due to heterogeneity within the stem/progenitor cell pool or the ability of similar stem/progenitor cells to adopt different fates stochastically. To date, most analysis of McSCs has relied on in vivo imaging and targeted transcriptomics, potentially missing the role of mixed transcriptional responses to injury. Single-cell RNA-sequencing has been used to dissect the extraordinary heterogeneity and transcription dynamics present in developing pigment cell lineages as well as melanocyte regeneration and offers new powerful insights into signaling mechanisms governing progenitor fate (*Infarinato et al., 2020*; *Saunders et al., 2019*).

Here, we utilize single-cell transcriptomics to find that zebrafish melanocyte progenitors are a heterogenous group of *mitfa*-expressing cells within the melanocyte stripe which require KIT signaling to directly differentiate during melanocyte regeneration. Importantly, changes in gene expression of progenitors during regeneration are well conserved between mice and zebrafish. Lastly, we observe

that heterogeneity of progenitors, evident in distinct transcriptional signatures, underlies the different fates adopted by these cells – direct differentiation versus division – during regeneration.

## Results

### Adult melanocyte lineage cells revealed by single-cell RNA-sequencing

During melanocyte regeneration, unpigmented *mitfa*-expressing cells in the zebrafish stripe engage in coupled differentiation and division to replenish lost melanocytes and maintain the pool of unpigmented progenitors (*Iyengar et al., 2015*). To identify the mechanisms governing these behaviors we sought to capture cell states and transcriptional changes in the melanocyte lineage during regeneration. Melanocyte progenitors are rare, so we generated a *Tg(mitfa:nlsEGFP)* reporter line and sorted EGFP-positive cells to isolate the 0.19% of total cells which express *mitfa* (*Figure 1A* and *Figure 1—figure supplement 1A*). We utilized neocuproine to ablate mature melanocytes, leading to the destruction of stripe melanocytes (*O'Reilly-Pol and Johnson, 2008*), then performed scRNAseq on *mitfa:nlsEGFP*-positive cells following melanocyte destruction (*Figure 1B*). In all, we obtained transcriptomes of 29,453 wild-type cells across six time points before, during, and after melanocyte regeneration. Quality control and pre-processing followed by dimensionality reduction and unsupervised clustering identified several subpopulations of cells (*Figure 1C*, *Supplementary file 1*). Analysis of captured cells revealed that 87.5% were positive for endogenous *mitfa* expression, reinforcing the success of our enrichment strategy (*Figure 1—figure supplement 1B*). The vast majority of *mitfa*-expressing cells were in the two large subgroups, and expression of *sox10*, *tyrp1b*, *aox5*, and other markers was consistent with these subgroups containing pigment cells (*Figure 1C, D*). Expression of the *tyrp1b* melanin biosynthesis gene was limited to one subpopulation, and we have assigned this subpopulation as differentiated (and differentiating) melanocytes. Expression of another differentiated melanocyte marker, *pmela*, confirmed this assignment (*Figure 1E*). The two large subgroups were distinguished from one another by their differential expression of the *aox5* gene (*Figure 1D*). *aox5* is expressed in differentiated xanthophores, a pteridine- and carotenoid-containing pigment cell type of zebrafish, and in some undifferentiated pigment progenitor cells (*McMenamin et al., 2014*; *Parichy et al., 2000*; *Saunders et al., 2019*). Expression of additional marker genes defined subpopulations of differentiated xanthophores and a third type of pigment cell, iridophores (*Figure 1—figure supplements 2A and 3A*). Additionally, *pcna* and *mki67* expression highlighted areas of proliferation within the *mitfa*+*aox5*$^{lo}$ and *mitfa*+*aox5*$^{hi}$ subgroups (*Figure 1—figure supplement 3B*). The non-*mitfa*-expressing clusters comprised other cell populations such as keratinocytes, fibroblasts, immune cells, and Schwann cells, based on expression of canonical cell-type marker genes (*Figure 1C, E*, *Figure 1—figure supplement 2A, B*, *Supplementary file 1*). Our cell-type assignments generally agree with cells obtained in a transcriptomic analysis of cell lineages that expressed the *sox10* neural crest and pigment cell marker during development, supporting the robustness of our approach to capture these cell types (*Figure 1—figure supplement 4*; *Saunders et al., 2019*).

### Single-cell transcriptomics reveal melanocyte progenitor heterogeneity and dynamic gene expression changes following melanocyte destruction

To investigate if progenitors were clustered in the same *mitfa*+*aox5*$^{lo}$ subgroup as melanocytes, we sought to understand the transcriptional and population changes of this subgroup during regeneration. Unsupervised clustering of the *mitfa*+*aox5*$^{lo}$ subgroup from integrated samples from the six time points before, during, and after melanocyte regeneration revealed four distinct subpopulations in addition to differentiated melanocytes (*Figure 2A*). Two of these subpopulations were characterized by low levels of *tyrp1b* expression and high levels of *sox4a* (*Figure 2B*, *Figure 2—figure supplement 1A*). SOX4 in mammalian cells has been shown in both normal and tumor cells to maintain an undifferentiated state that is associated with stemness (*Uy et al., 2015*; *Vervoort et al., 2013*). Because of this as well as the expression of genes typically observed in the neural crest (*Figure 2—figure supplement 1A, B*), the tissue from which melanocytes are derived during embryonic development, we considered these two subpopulations as putative progenitors and have designated them as melanocyte progenitor-0 (MP-0) and melanocyte progenitor-1 (MP-1). The MP-0 and MP-1 subpopulations themselves showed gene expression differences (e.g., *dio3a* and *bco2b*) that drove their separate

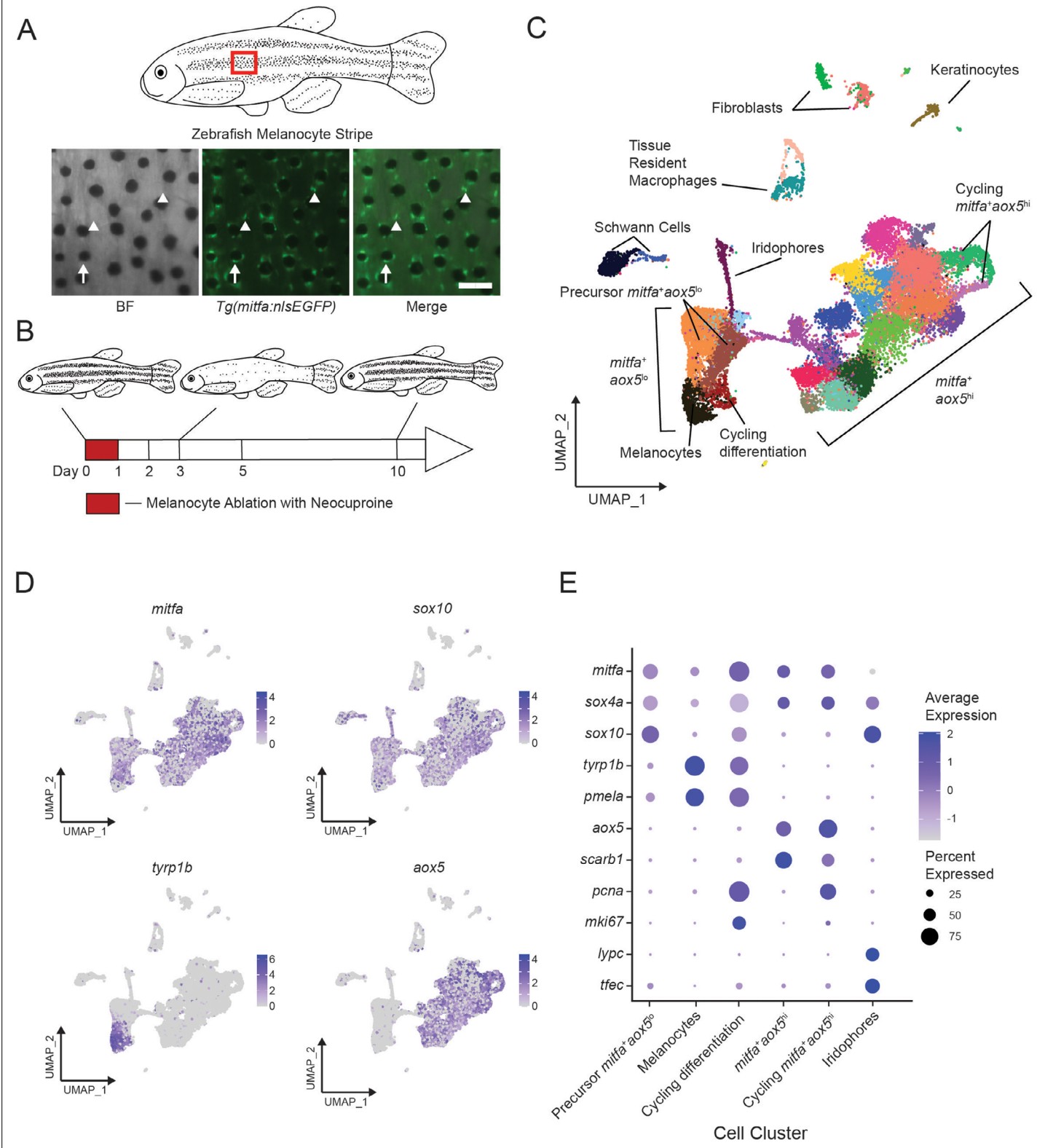

**Figure 1.** Single-cell transcriptomic identification of melanocyte lineage cells during regeneration. (**A**) Top, diagram of zebrafish flank with melanocyte stripes. Bottom, representative images of the melanocyte stripe in a *Tg(mitfa:nlsEGFP)* zebrafish. Melanocyte progenitors are unpigmented GFP-expressing cells (arrowheads) admixed with pigmented melanocytes (arrows). Animals were treated with epinephrine prior to imaging to concentrate melanosomes into the cell body of melanocytes. Scale bar = 100 μm. (**B**) Experimental design for transcriptional profiling of progenitors in

*Figure 1 continued on next page*

*Figure 1 continued*

*Tg(mitfa:nlsEGFP)* zebrafish during melanocyte ablation and regeneration. Cells from *Tg(mitfa:nlsEGFP)* zebrafish were sampled at the days specified. (**C**) UMAP of cell-type assignments for clusters of cells obtained from *Tg(mitfa:nls:EGFP)* zebrafish. Cells from all time points were included. Coloring is according to unsupervised clustering (*Blondel et al., 2008*; *Stuart et al., 2019*). Cells are labeled based on gene expression patterns revealed in panels (**D**) and (**E**) and *Figure 1—figure supplement 2A*. The large group of cells to the bottom left in the UMAP are *mitfa⁺aox5ˡᵒ*. Within this group of cells, the *mitfa⁺aox5ˡᵒtyrp1b⁺pcna⁻* cells are designated as melanocytes, the *mitfa⁺aox5ˡᵒtyrp1b⁺pcna⁺* are designated 'cycling differentiation', and the other two populations are designated 'precursor *mitfa⁺aox5ˡᵒ*'. The larger group of cells to the bottom right in the UMAP are *mitfa⁺aox5ʰⁱ*. Within this group of cells two clusters are *pcna⁺* are designated 'cycling *mitfa⁺aox5ʰⁱ* (n = 29,453 cells). (**D**) Expression of pigment cell markers *mitfa* and *aox5*, melanin biosynthesis gene *tyrp1b* and pigment cell progenitor marker *sox10* shown as feature plots on the UMAP plot from (**C**). (**E**) Expression of pigment cell markers, the stem cell gene *sox4a*, and cell cycle markers for cell clusters shown in panel C. Dot sizes represent percentage of cells in the cluster expressing the marker and coloring represents average expression.

The online version of this article includes the following source data and figure supplement(s) for figure 1:

**Figure supplement 1.** Isolation strategy and characteristics of *mitfa:nlsEGFP*-expressing cells.

**Figure supplement 1—source data 1.** Cell number and percentage of *mitfa*-expressing cells from FACS isolation, based on scRNAseq analyses.

**Figure supplement 2.** Isolation strategy and characteristics of *mitfa:nlsEGFP*-expressing cells.

**Figure supplement 2—source data 1.** Proportions of cell types across all *Tg(mitfa:nlsEGFP)* samples.

**Figure supplement 3.** Expression of xanthophore, iridophore, and cell cycle markers.

**Figure supplement 4.** Comparison of this article's and *Saunders et al., 2019* zebrafish pigment cell lineage datasets with expression of melanocyte, xanthophore, iridophore, and cell cycle markers.

cluster assignments (*Figure 2—figure supplement 1A, C*). The remaining three subpopulations expressed moderate to high levels of *tyrp1b* (*Figure 2B*). The subpopulation with highest *tyrp1b* was assigned as mature melanocytes, whereas the two subpopulations with moderate *tyrp1b* levels also expressed neural crest markers (*Figure 2—figure supplement 1A*), suggesting that they were not fully differentiated. To validate that our scRNAseq procedure was reproducible enough to enable comparisons between samples, we performed independent sampling of *mitfa*-positive cells without any melanocyte ablation. The cell sampling prevalence between single-cell runs was consistent, as seen by comparing proportions of cell types from two independent, pre-ablation samples (*Figure 2—figure supplement 1D*), giving confidence to our inter-sample comparisons. Comparisons of samples across different time points during regeneration compared to before melanocyte ablation (day 0) shows dynamic shifts in subpopulations following ablation and during regeneration (*Figure 2C, D*). As expected, the melanocyte subpopulation decreased substantially (by >95%) following neocuproine-mediated melanocyte destruction between days 0 and 1. The mature melanocyte subpopulation then increased in size as regeneration proceeded through day 10. Comparisons of subpopulations across regeneration also revealed changes in the two subpopulations that express neural crest markers and moderate levels of *tyrp1b* (*Figure 2—figure supplement 1A–C*). One of the intermediate subpopulations expressed high levels of cell cycle genes, including *pcna* and *cdk1* (*Figure 2B*, *Figure 2—figure supplement 1C*). We termed this subpopulation 'cycling differentiation'. The other intermediate population was also positionally between *tyrp1b*-negative and *tyrp1b*-positive cells but did not express cell cycle genes (*Figure 2B*, *Figure 2—figure supplement 1C*), so we termed this subpopulation 'direct differentiation'. Both the cycling differentiation and direct differentiation subpopulations grew following ablation, but then diminished as regeneration neared completion at day 10 (*Figure 2C, D*).

These differences in gene expression between subpopulations, and dynamic subpopulation sizes, suggested that cells transit from one subpopulation to another during regeneration. However, our analyses thus far were limited to our real-time sampling and could miss cellular transitions during biological process time. To interrogate whether transcriptional changes present in our samples reflect cellular transitions during regeneration, we utilized a latent variable, pseudotime analysis (*Trapnell et al., 2014*; *Wolf et al., 2018*). We applied the R package monocle3 to learn graph trajectories for our *mitfa⁺aox5ˡᵒ* subgroup of cells, which includes melanocyte progenitor and mature melanocyte subpopulations (*Cao et al., 2019*). We set the MP-0 subpopulation as pseudotime 0, an approach validated by known neural crest and melanocyte synthesis markers as well as RNA splicing mechanics as determined by RNA velocity analyses (*Figure 2—figure supplement 1E*; *Bergen et al., 2020*; *La Manno et al., 2018*). Pseudotime analysis projected onto the *mitfa⁺aox5ˡᵒ* subgroup of cells suggested two trajectories: one from the MP-0 subpopulation through the direct differentiation subpopulation

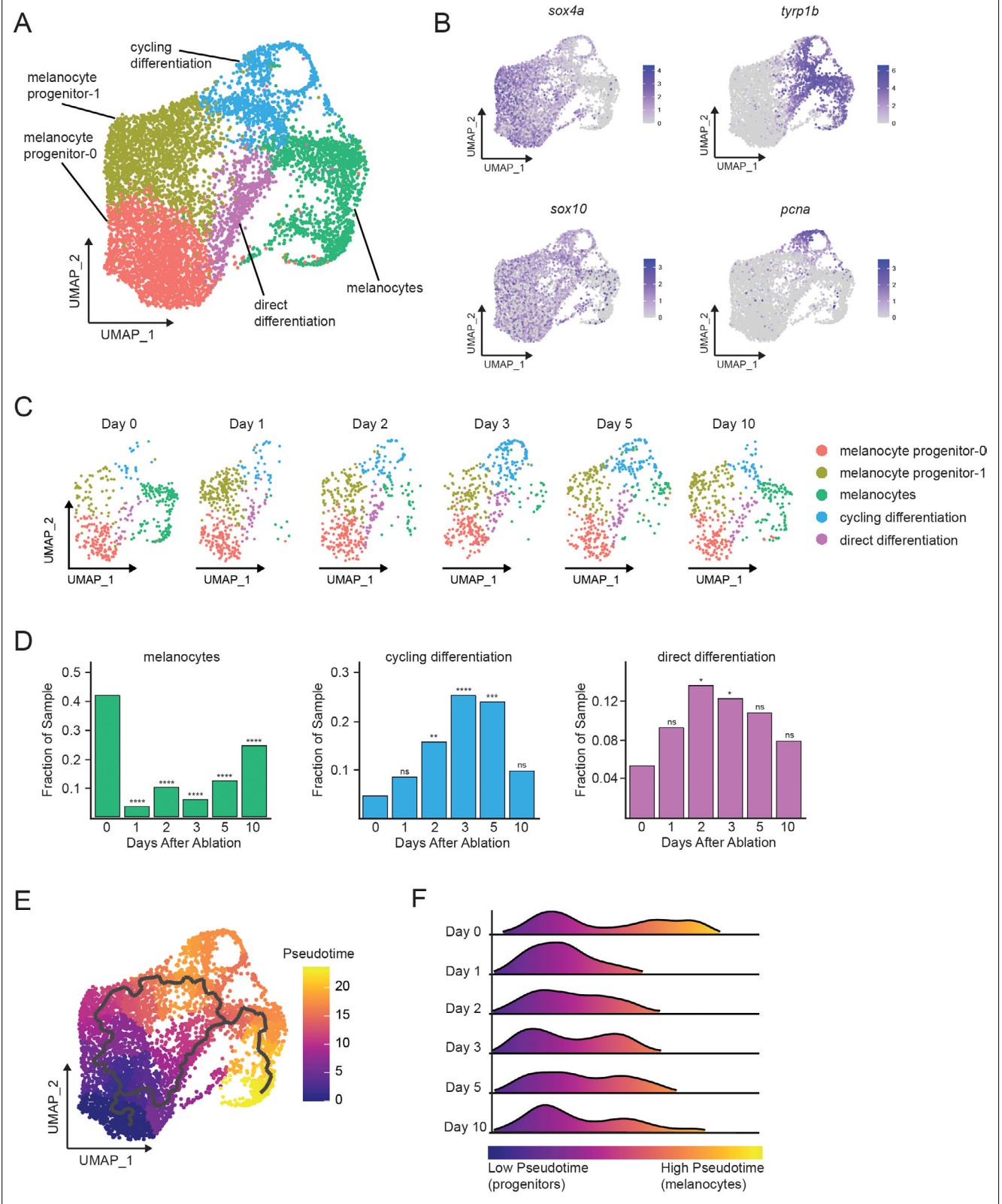

**Figure 2.** Single-cell transcriptomics reveal dynamic cell and gene expression changes following melanocyte ablation. (**A**) Integrated subclustering of WT *mitfa⁺aox5*ˡᵒ cells before, during, and after melanocyte regeneration (*n* = 5619 cells). Coloring is according to unsupervised clustering (*Blondel et al., 2008*; *Stuart et al., 2019*). Cell clusters are labeled based on gene expression patterns and inferred trajectories during regeneration, as revealed in panels (**B–F**) and *Figure 2—figure supplement 1*. Melanocyte and cycling differentiation clusters are the same as in *Figure 1A*, whereas

*Figure 2 continued on next page*

*Figure 2 continued*

the two precursor *mitfa⁺aox5*^lo clusters from ***Figure 1A*** are now resolved into three clusters: melanocyte progenitor-0 (MP-0), melanocyte progenitor-1, and 'direct differentiation', the latter of which expresses the stem cell marker *sox4a*, the melanin biosynthesis gene *tyrp1b* and is *pcna* negative. (**B**) Expression of stem cell marker *sox4a*, melanin biosynthesis gene *tyrp1b*, pigment cell progenitor marker *sox10*, and cell cycle gene *pcna* as feature plots on the UMAP plot from (**A**). (**C**) Dynamic changes in subpopulations of WT *mitfa⁺aox5*^lo cells. Biological sample runs were downsampled to a common number of total cells so shifts in cluster proportions could be readily visualized. (**D**) Quantification of proportion of cells per scRNAseq sample, comparing the indicated time point to day 0, in the melanocyte, cycling differentiation, and direct differentiation subpopulations during regeneration in WT animals (Day 0 = 7989, Day 1 = 4004, Day 2 = 4510, Day 3 = 5224, Day 5 = 3483, Day 10 = 4243 total cells per sample). p values calculated using differential proportion analysis (***Farbehi et al., 2019***), *p < 0.05; **p < 0.01; ***p < 0.001; ****p < 0.0001; ns, not significant. (**E**) Cellular trajectories, as determined by Monocle3 (***Cao et al., 2019***) projected onto the *mitfa⁺aox5*^lo subcluster (left panel). Solid lines represent trajectories, with an origin in the MP-0 subpopulation and two distinct paths through different intermediate cell subpopulations. (**F**) These trajectories are then used to calculate pseudotime, which, as an approximation of biological time, reveals how low pseudotime progenitors (blue) progress across transitional cell types to high pseudotime melanocytes (yellow). Ridge plot of the distribution of pseudotime during regeneration. Height of ridge corresponds to number of cells at that pseudotime.

The online version of this article includes the following source data and figure supplement(s) for figure 2:

**Source data 1.** Proportion of cells per scRNAseq sample in the melanocyte, cycling differentiation, and direct differentiation subpopulations.

**Figure supplement 1.** Single-cell profiling supports progenitor identity and subpopulation dynamics.

**Figure supplement 1—source data 1.** Comparison of *mitfa⁺aox5*^lo cell subpopulation proportions between scRNAseq samples of unperturbed (day 0) wild-type zebrafish skin.

into melanocytes and another branching through the MP-1 and cycling differentiation subpopulations into melanocytes (***Figure 2E***). Using this graph projection, we ordered the cells by biological pseudo-time (***Figure 2F***). Progenitors are low pseudotime cells (enriched for *sox4a*, *sox10*, and no melanin synthesis genes) while melanocytes are high pseudotime cells (enriched for *tyrp1b* and *pmela*). High pseudotime melanocytes were lost following ablation between days 0 and 1, resulting in a distri-bution of predominantly low pseudotime progenitors at day 1. Then, as regeneration proceeded the intermediate cell states became enriched and can be visualized as medium pseudotime cells in days 2–5. Ultimately, as regeneration proceeded, high pseudotime melanocytes were regained. This pseudotime dynamic mirrors what is seen in vivo when unpigmented *mitfa*-positive progenitor cells divide or directly differentiate, reinforcing that the progenitor populations we have defined through scRNAseq correspond to progenitors observed in vivo. Together these analyses provide a compre-hensive picture of changes in progenitor transcriptomes during melanocyte regeneration which mirror real-time regeneration observations.

To assess if our approach could inform broader mechanisms governing melanocyte regeneration, we compared our zebrafish melanocyte progenitor signatures with a mammalian McSC signature. A single-cell transcriptomic analysis of murine progenitors during hair follicle homeostatic cycling provided an opportunity to compare zebrafish and mammalian signatures (***Infarinato et al., 2020***). Clustering of 626 wild-type murine McSCs and melanocytes revealed four populations of previously described cells: quiescent McSCs (qMcSCs), activated McSCs (aMcSCs), cycling McSCs, and mela-nocytes (***Figure 3—figure supplement 1A***; ***Infarinato et al., 2020***). Visualization of this structure reveals similarities to the previously computed zebrafish clustering, where non-cycling progenitors are separated from mature melanocytes by intermediate cell populations. To assess similarities in gene expression we calculated marker genes for each zebrafish and murine subpopulation, and then visualized conserved gene signatures via heatmaps (***Figure 3—figure supplement 1B***). As murine McSCs progress from qMcSCs to differentiated melanocytes they lose expression of *Zeb2*, *Pax3*, and AP-1 FOS and JUN family subunits, while upregulating melanin synthesis genes such as *Tyrp1* and *Oca2*. These changes mirror the transcriptional profiles found in our zebrafish dataset. Lastly, to visualize correlation between zebrafish and murine populations we calculated cluster-specific differ-entially expressed genes (DEGs), mapped these to murine orthologs, and then scored the murine populations with these zebrafish signatures (***Figure 3—figure supplement 1C***, ***Supplementary file 2***). Through these comparisons, we find that our MP-0 and MP-1 subpopulations are most similar to qMcSCs. The zebrafish direct differentiation subpopulation, much like when it is compared to other zebrafish subpopulations, shares aspects of murine McSC and more differentiated subpopulations. The zebrafish cycling differentiation subpopulation most strongly overlaps with murine cycling McSC subpopulation. And as expected, mature pigment producing melanocytes in each dataset are most

like each other. These signature scores reflect the positional similarities we see in UMAP visualization where qMcSC and MP-0 and MP-1 are most stem-like and least differentiated. Overall, these signatures support a shared gene signature of differentiation during melanocyte progenitor activation, supporting the use of zebrafish to uncover conserved signaling pathways governing melanocyte regeneration.

## Signaling pathway dynamics in progenitors uncover the KIT signaling axis as a candidate regulator of melanocyte regeneration

To identify signaling pathways governing melanocyte regeneration we first analyzed potential receptor–ligand interactions using NicheNetR (*Browaeys et al., 2020*). In brief, NicheNetR calculates transcriptional changes in a target 'receiver' cell population between time samples. It uses these DEGs to predict ligands responsible for the observed behavior. Finally, it matches the expression of these ligands on potential 'sender' cells with cognate receptors on the receiver cell subpopulation. We designated our identified progenitor subpopulations as receptor-expressing receiver cells and all other cell types as potential ligand sender cells. We compared gene expression in our regenerating progenitors (MP0 and MP1) on days 2, 3, and 5 to progenitor gene expression in the unactivated day 0 controls. Data shown are from day 3 versus day 0, although comparisons of day 2 or 5 versus day 0 show similar results. We then filtered down top scoring ligand/receptor pairs to NicheNetR's 'bona fide' literature-supported pairs. This approach identified several signaling systems with known roles in melanocyte biology, including KITLG/KIT, ASIP/MC1R, EDN3/EDNRB, and NRG1/ERBB2 (*Figure 3A*; *Chou et al., 2013*; *Hultman et al., 2009*; *Li et al., 2017*; *Yamada et al., 2013*). These signaling systems function during melanocyte development, and our data indicate they are also reactivated during melanocyte regeneration. To more deeply understand how such reactivation would modulate regeneration, we focused on KITLG/KIT signaling. In zebrafish the KIT receptor ortholog, *kita*, is necessary for the survival and migration of melanocytes as they emerge from the neural crest (*Rawls and Johnson, 2003*) and has been implicated in establishment of larval progenitors (*O'Reilly-Pol and Johnson, 2013*; *Yang et al., 2004*), but whether the KITLG/KIT pathway is reactivated and how it might govern progenitor fates during regeneration have not been addressed.

We analyzed expression of *kita* and its ligand *kitlga* during regeneration. *kita* receptor was expressed in melanocyte progenitor, cycling differentiation and direct differentiation subpopulations (*Figure 3B*). To test for dynamism in the KIT signaling axis, we sought to assay *kita* receptor and *kitlga* ligand expression following melanocyte destruction (*Figure 3C*). From bulk skin samples, levels of *kita* receptor did not change dramatically during regeneration. By contrast, *kitlga* ligand levels changed such that there was an increase of *kitlga* expression shortly following melanocyte destruction, which then tapered downwards as regeneration proceeded. We analyzed scRNAseq data to determine cell types that express *kitlga*. Thanks to trace numbers of *mitfa*-negative cells that were analyzed as part of our scRNAseq, we were able to observe *kitlga* expression in fibroblasts and keratinocytes, mirroring scRNAseq expression patterns from human cells (*Figure 3—figure supplement 2A, B*; *Joost et al., 2020*). Previous studies have shown that KIT has a critical role in mediating lineage decisions in melanocyte development as well as melanocyte homeostasis in the murine hair follicle (*Botchkareva et al., 2003*; *Botchkareva et al., 2001*; *Geissler et al., 1988*; *Rawls and Johnson, 2003*). KITLG/KIT signaling also has known roles in human melanoma as well as murine and zebrafish melanoma models (*Beadling et al., 2008*; *Santoriello et al., 2010*). Our data indicate that KITLG/KIT signaling is reactivated in response to injury, making this signaling axis a leading candidate as a regulator of melanocyte progenitor fates during regeneration.

## kita receptor and kit ligand loss-of-function mutants impair direct differentiation of progenitors during melanocyte regeneration

To directly test the role of KITLG/KIT signaling predicted by transcriptomics we measured melanocyte regeneration in *kita(lf)* and *kitlga(lf)* animals. Whereas wild-type animals regenerated their full contingent of stripe melanocytes within 2 weeks of neocuproine-induced melanocyte destruction, *kita(lf)* mutants regenerated a mean of 51.4% of their melanocytes indicating a requirement for KIT signaling during regeneration (*Figure 4A, B*). *kitlga(lf)* mutants were similarly defective in regeneration (*Figure 4A, B*). Previous work established that zebrafish regeneration melanocytes arise from one of two sources: direct differentiation of progenitors or, less frequently, asymmetric divisions of

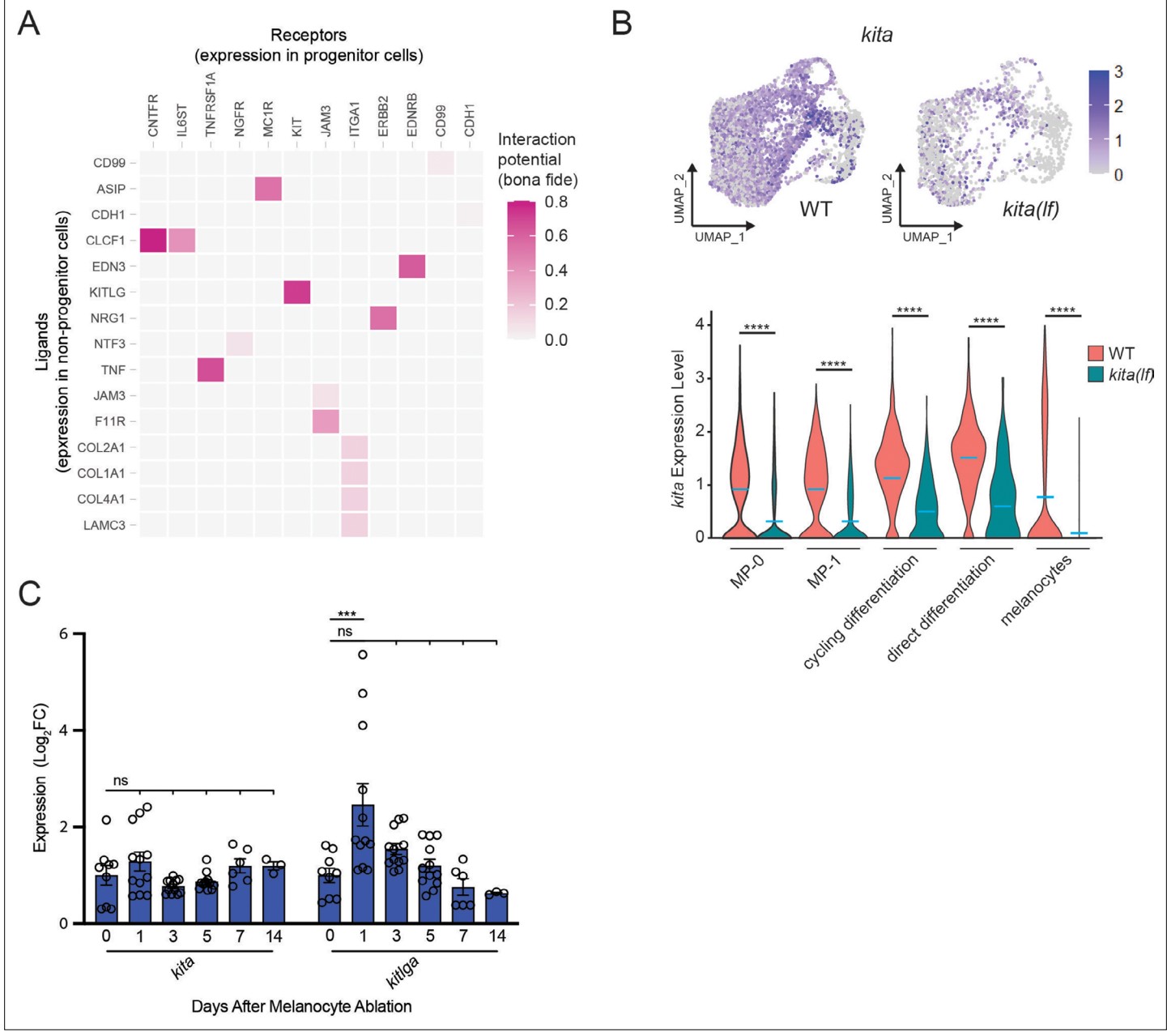

**Figure 3.** NicheNet and transcriptional analyses implicate the KIT signaling axis as a dynamic regulator of melanocyte regeneration. (**A**) Heatmap of NicheNetR-identified ligand/receptor pairs indicating interaction potential between receptors on progenitor receiver cells and ligands on non-progenitor sender cells sampled by scRNAseq. Ligand/receptor pairs were restricted to literature-supported pairs (**Browaeys et al., 2020**). (**B**) Top, feature and, bottom, violin plots of *kita* expression in *mitfa⁺aox5^lo* cells from wild-type (MP-0 = 1840, MP-1 = 1408, cycling differentiation = 777, direct differentiation = 535, and melanocytes = 1060 cells) and *kita(lf)* strains (MP-0 = 617, MP-1 = 612, cycling differentiation = 280, direct differentiation = 57, and melanocytes = 563 cells). Mean gene expression represented by cyan bars. p values calculated by Wilcoxon rank-sum test, ****p < 0.0001. (**C**) Quantitative real-time polymerase chain reaction (qRT-PCR) of *kita* and *kitlga* expression in zebrafish skin following melanocyte destruction. Three biological replicates were performed for each time point. Data are shown as mean ± standard error of the mean (SEM). p values calculated by one-way analysis of variance (ANOVA) with Dunnett's multiple comparisons test, ***p < 0.001; ns, not significant.

The online version of this article includes the following source data and figure supplement(s) for figure 3:

**Source data 1.** qRT-PCR of *kita* and *kitlga* expression in zebrafish skin following melanocyte destruction.

**Figure supplement 1.** Zebrafish progenitor regenerative signature is conserved across species.

**Figure supplement 2.** *kitlga* ligand expression in zebrafish keratinocytes and fibroblasts is similar to that observed in human cells.

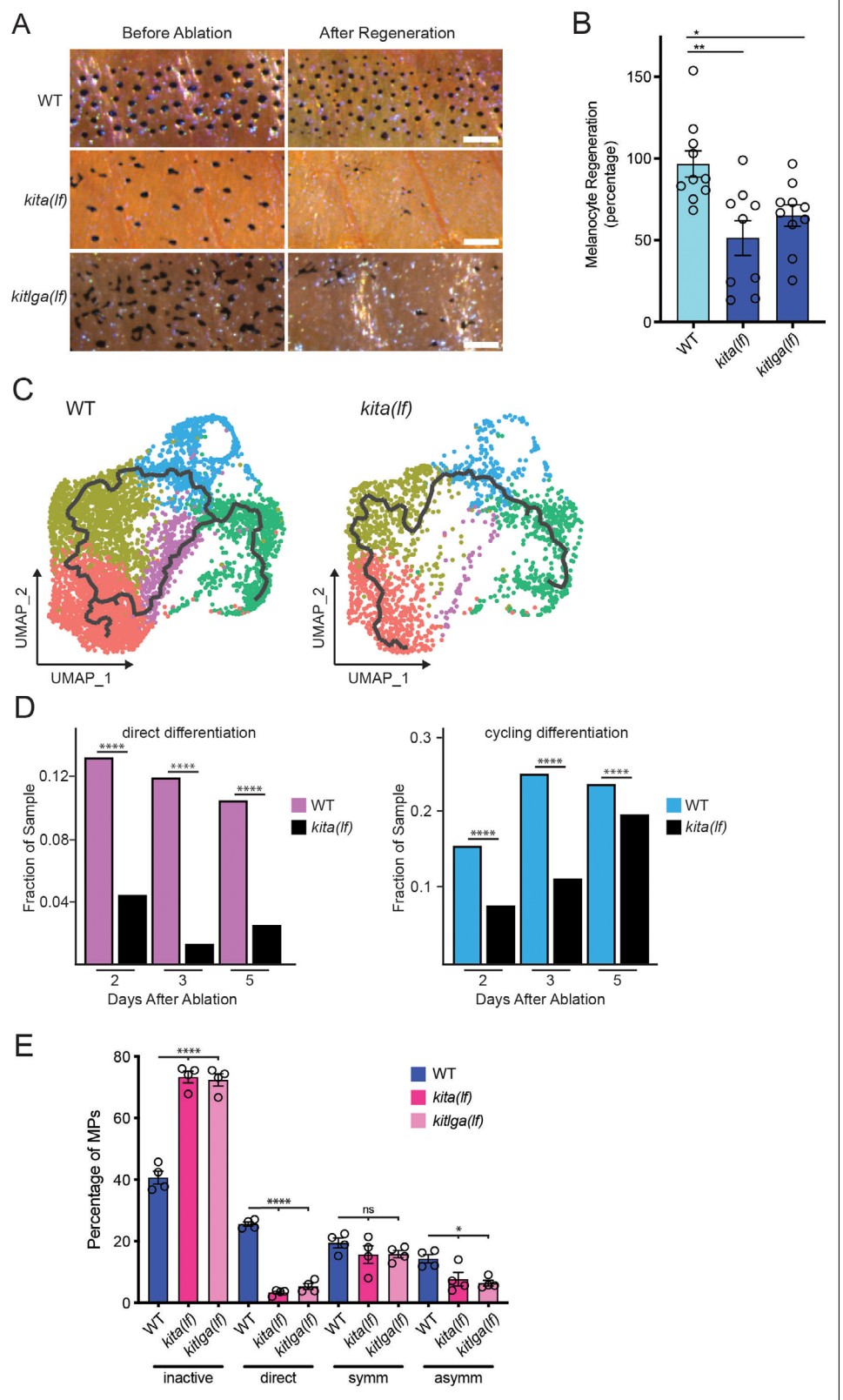

**Figure 4.** *kita* receptor and *kitlga* ligand loss-of-function mutants have impaired melanocyte regeneration. (**A**) Brightfield images of the melanocyte stripe before melanocyte ablation and after melanocyte regeneration in wild-type, *kita(lf)*, and *kitlga(lf)* zebrafish strains. Scale bar = 200 μm. (**B**) Quantification of melanocyte regeneration in wild-type, *kita(lf)*, and *kitlga(lf)* strains. Mean percentage ± standard error of the mean (SEM) is shown; WT *n*

*Figure 4 continued on next page*

*Figure 4 continued*

= 10, *kita(lf)* = 9, *kitlga(lf)* = 11 fish. p values calculated by one-way analysis of variance (ANOVA) with Dunnett's multiple comparisons test, *p < 0.05, **p < 0.01. (**C**) Cellular trajectories, using Monocle3, of *mitfa*+*aox5*lo cells from wild-type zebrafish (left) and *kita(lf)* mutants (right). Solid lines represent trajectories, with an origin in the melanocyte progenitor-0 (MP-0) subpopulation and terminus in the melanocyte subpopulation. (**D**) Comparison of proportion of WT and *kita(lf)* cells per scRNAseq sample in the cycling differentiation and direct differentiation subpopulations during regeneration reveals fewer *kita(lf)* cells going through differentiation (WT Day 2 = 4510, WT Day 3 = 5224, WT Day 5 = 3483, *kita(lf)* Day 2 = 4233, *kita(lf)* Day 3 = 5261, *kita(lf)* Day 5 = 5411 total cells per sample). p values calculated using differential proportion analysis (*Farbehi et al., 2019*), ****p < 0.0001; ns, not significant. (**E**) Fates of progenitors following single-cell serial imaging of wild-type, *kita(lf)*, and *kitlga(lf)* strains. Mean percentage of traced progenitors in a fate are shown; wild-type *n* = 4 animals (*n* = 49, 59, 53, 53 cells per animal), *kita(lf)* = 4 animals (*n* = 50, 50, 53, 28 cells per animal), *kitlga(lf)* = 4 animals (*n* = 52, 39, 53, 33 cells per animal). p values calculated by one-way ANOVA with Dunnett's multiple comparisons test, ****p < 0.0001, *p < 0.05, ns, not significant.

The online version of this article includes the following source data and figure supplement(s) for figure 4:

**Source data 1.** KIT signaling mutants have impaired melanocyte regeneration.

**Figure supplement 1.** *kita(lf)* animals demonstrate conservation of cell types found in WT animals (clustered together).

**Figure supplement 2.** *kita(lf)* animals demonstrate conservation of cell types found in WT animals (clustered separately).

---

progenitors in which one of two daughter cells undergoes differentiation (*Iyengar et al., 2015*). To determine whether one or both progenitor fates were defective in *kita* mutants we sequenced 24,724 transcriptomes from *mitfa:nlsEGFP*-sorted cells from regenerating *kita(lf)* mutants. Integrated clustering of cells from *kita(lf)* mutant and wild-type strains revealed conservation of the cell types found in both strains (*Figure 4—figure supplement 1*, *Figure 4—figure supplement 2*). Additionally, analysis of *kita* expression showed that *kita(lf)* mutants did indeed express lower levels of *kita* transcripts, likely due to nonsense-mediated decay tied to a premature stop codon encountered after the frameshift mutation in the *kita(b5)* allele used in our study (*Figure 3B*; *Parichy et al., 1999*). We analyzed scRNAseq data and performed in vivo serial imaging to determine the basis for the regeneration defect in *kita/kitlga* animals. Trajectory analysis using monocle3 revealed that the pathway from progenitor subpopulations through the direct differentiation subpopulation to melanocytes was absent in *kita(lf)* mutants (*Figure 4C*). This absence stemmed from the lack of an increase in the direct differentiation subpopulation during regeneration (*Figure 4D*). The cycling differentiation subpopulation was also affected, with a smaller increase during regeneration observed as compared to the wild-type. To assess if these differences resulted from lineage defects during regeneration, we performed serial imaging in *kita(lf)* mutants containing the *mitfa(nls:EGFP)* transgene. Serial imaging revealed a greater than eightfold decrease in progenitors undergoing direct differentiation (*Figure 4E*). Asymmetric divisions were also decreased by twofold. Serial imaging of *kitlga(lf); Tg(mitfa:nlsEGFP)* mutants showed similar defects as those observed in *kita(lf)* mutants (*Figure 4E*). Together, these data elucidate paths and signaling requirements for the two fates adopted by progenitors to regenerate new melanocytes. In one fate progenitors directly differentiate through an intermediate cell state before becoming melanocytes. This fate has a strong requirement for KIT signaling. In the other fate, progenitors take part in asymmetric divisions to generate new melanocytes. This fate path is characterized by co-expression of differentiation and cell cycle genes and is also affected, albeit less so, by defects in KIT signaling.

## Kit-mediated differentiation depends on MAPK pathway activity

The defect in regeneration caused by loss of *kita* receptor or *kitlga* ligand suggests that signaling downstream of the KIT receptor is required for proper melanocyte progenitor differentiation during regeneration. One well-documented downstream pathway is the MAPK pathway, with ERK being a known controller of the melanocyte lineage master regulator *mitfa* (*Hemesath et al., 1998*; *Levy et al., 2006*; *Wellbrock and Arozarena, 2015*; *Wellbrock et al., 2008*). Accordingly, we hypothesized that if KITLG/KIT signaling is important in regeneration, then ERK activity would be upregulated in melanocyte progenitors following melanocyte destruction. To test this hypothesis, we utilized in

vivo imaging with a kinase translocation reporter, ERKKTR-mClover (*Mayr et al., 2018*; *Regot et al., 2014*). In this system ERK activation via phosphorylation drives the ERKKTR-mClover fusion protein from a nuclear to cytosolic localization. These shifts in subcellular localization allow quantification of ERK activity and, by extension, MAPK pathway activity. We generated a MAPK activity reporter that drives ERKKTR-mClover expression from a melanocyte lineage-specific *mitfa* promoter and injected this reporter construct into a *Tg(mitfa:nlsmCherry)* strain to enable accurate measurement of nuclear/cytosolic intensity. We first observed that progenitors showed lower ERK activity than mature melanocytes (*Figure 5A, B*), indicating that ERK activity in progenitors is relatively low under homeostatic conditions.

To further understand the role of MAPK signaling in regeneration we assayed ERKKTR-mClover localization following melanocyte ablation. Following ablation, ERKKTR-mClover signal in wild-type progenitors shifted to an increased cytosolic localization, indicative of an increase in MAPK activity during regeneration (*Figure 5C, D*). Next, to test whether this increase was dependent on KITLG/KIT signaling, we injected the *mitfa:ERKKTR-mClover* reporter into a *kita(lf); Tg(mitfa:nlsmCherry)* strain. Progenitors in *kita(lf)* mutants displayed similar levels of ERK activity in homeostatic conditions as compared to progenitors in wild-type animals (*Figure 5C, D*). However, following melanocyte injury the *kita(lf)* animals showed no cytosolic translocation of ERKKTR-mClover, indicating a failure to upregulate MAPK activity in the absence of KIT signaling (*Figure 5C, D*). These results indicate that progenitors upregulate MAPK activity during regeneration, and blockade of KIT signaling impedes this upregulation.

If KIT-mediated MAPK signaling is required for direct differentiation during regeneration, then rescue of downstream MAPK signaling in *kita(lf)* mutants should result in rescue of the regeneration phenotype. To test this hypothesis, we utilized a constitutively active RAF, BRAF$^{V600E}$, a component of the MAPK pathway downstream of KIT but upstream of ERK (*Davies et al., 2002*; *Patton et al., 2005*). Introduction of the overactive BRAF had little effect in wild-type animals, but rescued regeneration in *kita(lf)* animals resulting in a melanocyte density that was greater than that of unperturbed *kita(lf)* animals and more comparable to that of wild-type animals (*Figure 5E, F*; *Figure 5—figure supplement 1*). This robust repigmentation suggests that adult melanocyte progenitors are more sensitive to RAS/MAPK signaling, via constitutively activated BRAF, than their ontogenetic counterparts. Overall, these results show that functional KIT signaling through the MAPK pathway is necessary and sufficient for proper melanocyte regeneration.

## scRNAseq identifies *mitfa⁺aox5$^{hi}$* cells that undergo divisions following melanocyte destruction

Multiple rounds of melanocyte injury can be performed in zebrafish without diminishing the capacity to regenerate. This is due to a subset of progenitors that divide symmetrically to maintain the pool of progenitors. Previous studies found that daughters of cells that divide during one round of regeneration are capable of melanocyte differentiation following subsequent injury, indicating a potential link between self-renewal and eventual differentiation (*Iyengar et al., 2015*). Given their central role in melanocyte regeneration, we sought to understand characteristics of progenitors that undergo symmetric divisions through our scRNAseq dataset.

To begin, we investigated our scRNAseq dataset for cells that entered the cell cycle in response to melanocyte injury. As previously mentioned, a subpopulation of *mitfa⁺aox5$^{lo}$* cells enter the cell cycle and express differentiation markers. Trajectory analysis indicates some of these cells ultimately differentiate into melanocytes, suggesting that the dividing cells may be ones that undergo asymmetric divisions and generate a differentiated daughter cell during regeneration. In addition to these cells, we identified separate *mitfa⁺aox5$^{hi}$* cycling cells that formed a characteristic loop in clustering analysis. Cells in one part of this loop were *pcna*-enriched and high in S phase score (*mitfa⁺aox5$^{hi}$* S phase) and cells in another part were *cdk1*-enriched and high in G2/M score (*mitfa⁺aox5$^{hi}$* G2/M phase) (*Figure 6A, B*, *Figure 6—figure supplement 1A*). Neither of these two subpopulations expressed the melanization genes *tyrp1b* and *pmela* (*Figure 6—figure supplement 1B*). Plotting these two subpopulations from each sampled time point revealed very few cells prior to melanocyte injury (day 0), a robust increase during regeneration (days 1, 2, 3, and 5) and a diminution when regeneration was mostly complete (day 10) (*Figure 6C, D*). The cell cycle phase scoring indicated a clockwise cellular trajectory from *mitfa⁺aox5$^{hi}$* S phase to *mitfa⁺aox5$^{hi}$* G2/M phase subpopulations, and UMAP

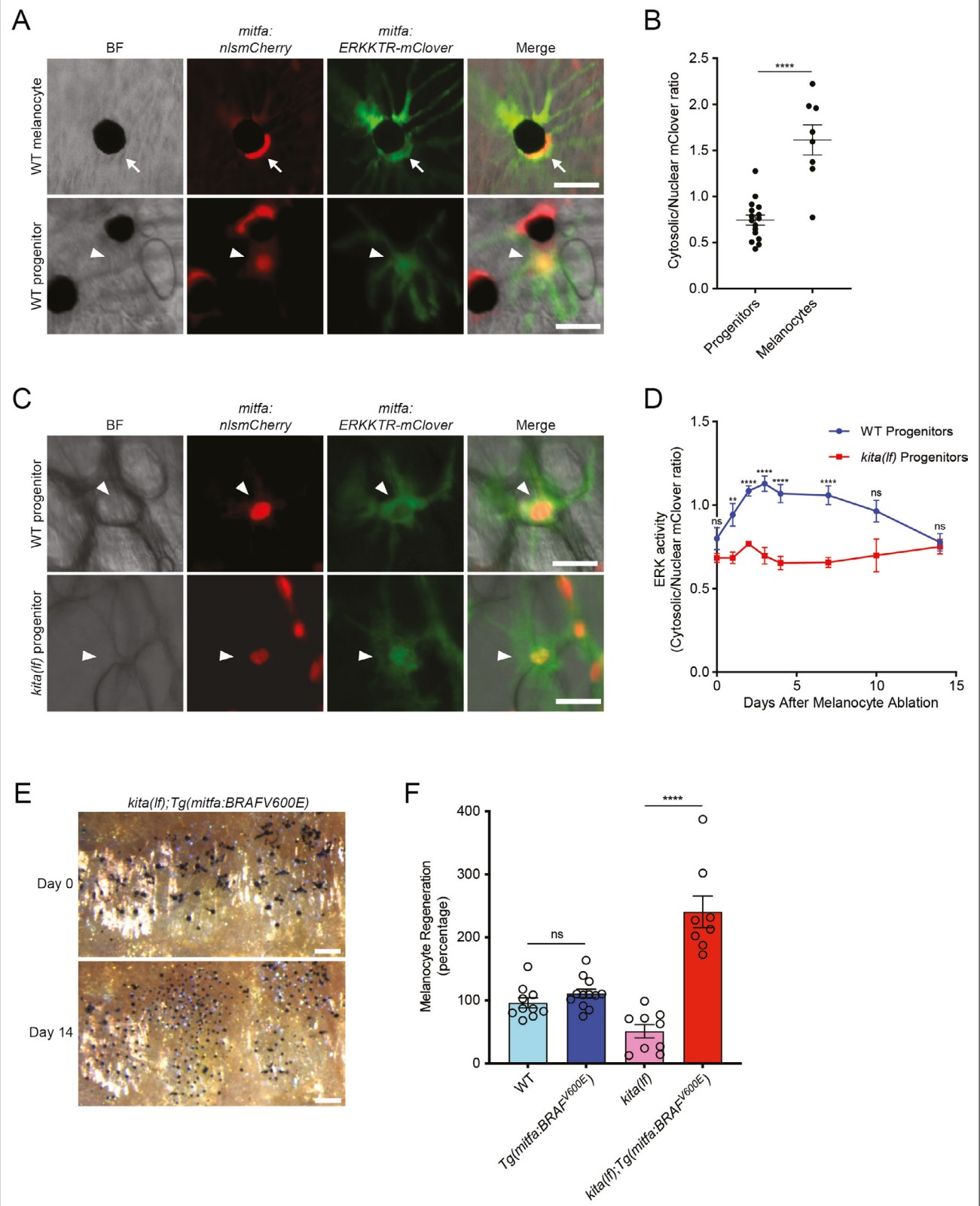

**Figure 5.** *kitlga/kita* signaling during melanocyte regeneration acts through the MAPK pathway. (**A**) Images of ERKKTR-mClover localization in a representative mature melanocyte (top, arrow) and progenitor (bottom, arrowhead) in uninjured wild-type animals. Scale bar = 30 µm. (**B**) Quantification of ERK activity in progenitors and melanocytes based on ERKKTR-mClover localization. Mean ± standard error of the mean (SEM) is shown; progenitors *n* = 16, melanocytes *n* = 8. (**C**) Images 3 days post-ablation of ERKKTR-mClover location in representative progenitors (arrowheads) in wild-type

*Figure 5 continued*

(top) and *kita(lf)* (bottom) animals. Scale bar = 30 μm. (**D**) Quantification of ERK activity in progenitors prior to and during melanocyte regeneration. For each data point, the average cytosolic/nuclear ratio of at least 6 cells ± SEM is shown. (**E**) Brightfield images of the melanocyte stripe before (top) and after (bottom) regeneration in *kita(lf); Tg(mitfa:BRAFV600E)* mutants. Scale bar = 200 μm. (**F**) Quantification of melanocyte regeneration in *kita(lf);Tg(mitfa:BRAFV600E)* and control animals. Mean percentage ± SEM is shown; wild-type *n* = 10, *Tg(mitfa:BRAFV600E)* = 12, *kita(lf)* = 9, *kita(lf);Tg(mitfa:BRAFV600E)* = 8 fish. p values calculated by Student's *t*-test, \*\*p < 0.01, \*\*\*\*p < 0.0001; ns, not significant.

The online version of this article includes the following source data and figure supplement(s) for figure 5:

**Source data 1.** KIT signaling during melanocyte regeneration acts through the MAPK pathway.

**Figure supplement 1.** *kita(lf)* animals with constitutively active BRAF regenerate a similar number of melanocytes as wild-type animals.

**Figure supplement 1—source data 1.** Number of melanocytes from the middle stripe per field in wild-type, *Tg(mitfa:BRAFV600E)*, *kita(lf)*, and *kita(lf); Tg(mitfa:BRAFV600E)* animals.

projection suggested that an adjacent subpopulation (*mitfa⁺aox5ʰⁱ* S adj) was an input to *mitfa⁺aox5ʰⁱ* S phase and a separate adjacent subpopulation (*mitfa⁺aox5ʰⁱ* G2/M adj) was an output from *mitfa⁺aox5ʰⁱ* G2/M phase (*Figure 6A*). As was evident from their separate clustering, *mitfa⁺aox5ʰⁱ* S adj and *mitfa⁺aox5ʰⁱ* G2/M adj subpopulations had differential expression of several genes, and among genes differentially upregulated in the *mitfa⁺aox5ʰⁱ* G2/M adj subpopulation were *foxd3*, *zeb2a*, and *vim*, which are associated with multipotent pigment progenitor cells (*Figure 6E*; *Budi et al., 2011*; *Saunders et al., 2019*). Furthermore, *sox4a* was differentially upregulated in the *mitfa⁺aox5ʰⁱ* G2/M adj subpopulation, suggesting a possible relationship with MP-0 and MP-1 subpopulations (*Figure 6E*).

The dynamics of cell division and absence of melanocyte differentiation genes make the *mitfa⁺aox5ʰⁱ* G2/M adj subpopulation a candidate for cells that undergo symmetric divisions during regeneration. However, because of their expression of *aox5* and other markers associated with xanthophores, we considered the possibility that these *mitfa⁺aox5ʰⁱ* cycling cells were xanthophore lineage cells cycling in response to ablation of mature xanthophores by neocuproine. To investigate this possibility, we traced inter-stripe xanthophores prior to and following neocuproine treatment. Consistent with previous reports (*O'Reilly-Pol and Johnson, 2008*) we observed no xanthophore ablation caused by neocuproine, and out of 65 inter-stripe xanthophores traced following neocuproine ablation we observed zero cell divisions (*Figure 7—figure supplement 1*). In the absence of xanthophore ablation and regeneration, we directly tested the hypothesis that the symmetrically dividing cells in the melanocyte stripe were *mitfa⁺aox5ʰⁱ* cells evident in scRNAseq profiling. We combined our *mitfa:nlsm-Cherry* reporter with an *aox5:PALM-EGFP* reporter (*Eom et al., 2015*) and investigated dual-positive cells in the melanocyte stripe. All of the unpigmented *mitfa*-positive cells in the melanocyte stripe also expressed *aox5:PALM-EGFP*. The dual-positive cells exhibited heterogeneity in their *aox5:PALM-EGFP* expression consistent with one subpopulation being *aox5ʰⁱ* and another subpopulation *aox5ˡᵒ* (*Figure 7A, B*). To determine if *mitfa⁺aox5ʰⁱ* cells correspond to cells that undergo symmetric divisions during regeneration, we tracked dual-positive cells following melanocyte ablation (*Figure 7C*). Cells that underwent symmetric divisions during regeneration exhibited high *aox5⁺* before melanocyte regeneration (*Figure 7D*). Taken together, these results suggest that the cycling *mitfa⁺aox5ʰⁱ* cells from our scRNAseq correspond to pigment lineage cells in the melanocyte stripe that undergo symmetric divisions following melanocyte destruction.

## Discussion

Stem and progenitor cells that are activated in tissue regeneration and homeostasis have the capacity to generate newly differentiated cells and maintain a pool of stem cells. In this study, we combined scRNAseq with single-cell serial imaging and other analyses to determine how this occurs during regeneration of melanocyte stripes in zebrafish. Interestingly, we found that progenitor heterogeneity and fate-specific signaling systems underlie a coordinated regeneration process.

Progenitor heterogeneity is illustrated, in part, by the two subpopulations of *mitfa⁺aox5ˡᵒ* progenitors, MP-0 and MP-1. MP-0 and MP-1 are transcriptionally distinct and are present in unperturbed animals. RNA velocity and cell trajectory analyses indicate that these subpopulations are activated upon melanocyte ablation and that each supplies different intermediate subpopulations during the regenerative process. MP-1 cells feed primarily into the cycling differentiation subpopulation, whereas MP-0 cells feed into the direct differentiation subpopulation. Additionally, these analyses suggest

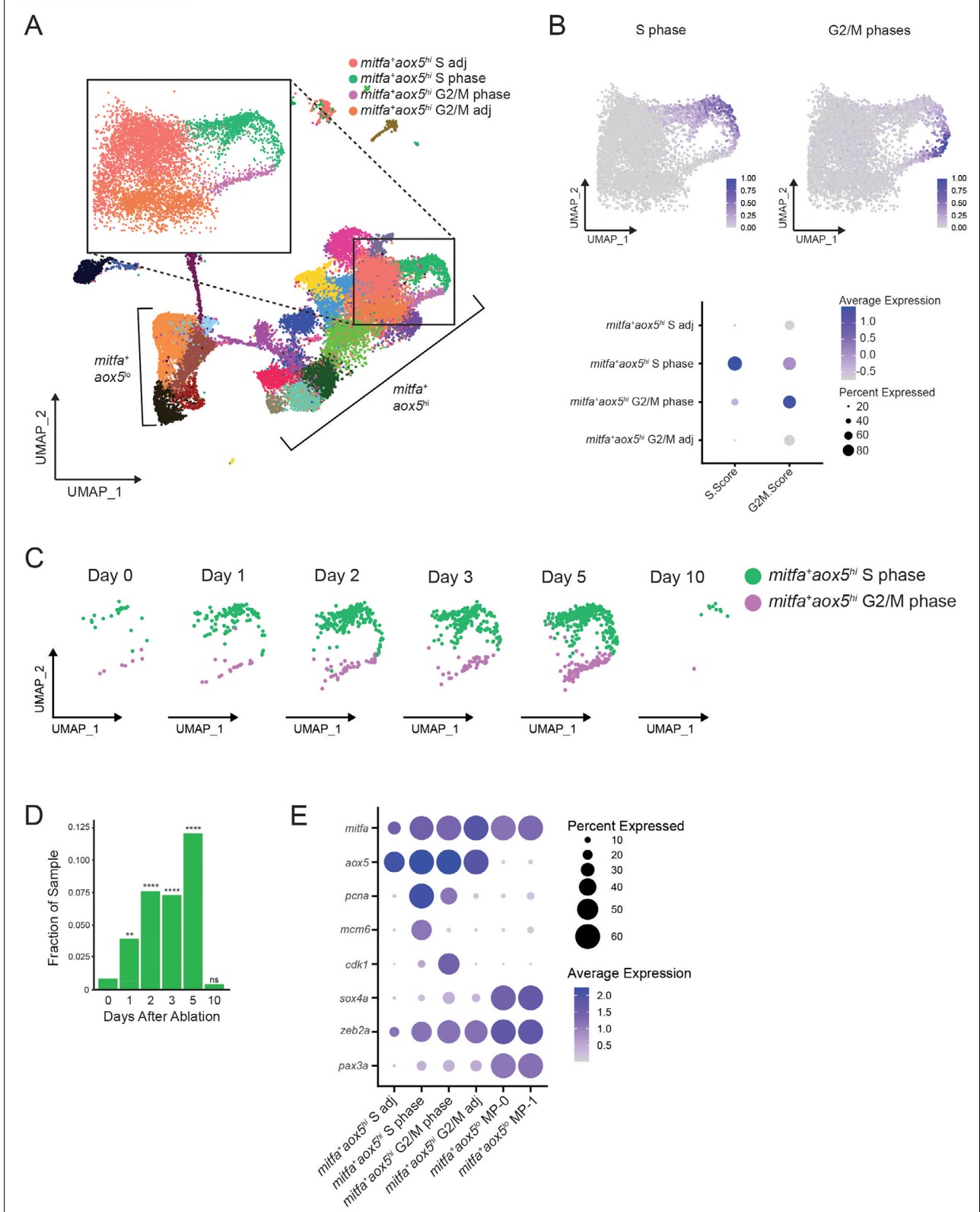

**Figure 6.** A subpopulation of *mitfa+aox5*hi cells divides and expands during regeneration. (**A**) UMAP of all cells sampled from wild-type animals with enlargement of cycling and adjacent subpopulations found in the large group of *mitfa+aox5*hi cells. (**B**) Top, feature and, bottom, dot plot of S phase and G2/M phase cell cycle scores of cells highlighted in (**A**). Cell cycle scores were calculated using Seurat's 'CellCycleScoring' module and zebrafish orthologs of the cell cycle genes outlined by *Tirosh et al., 2016*. (**C**) UMAPs of cycling cells prior to and during regeneration. Biological sample runs

*Figure 6 continued on next page*

*Figure 6 continued*

were downsampled to a common number of total cells so shifts in clusters could be readily visualized. Cells in S phase are seen in green, and cells in G2/M phase are seen in purple. (**D**) Quantification of proportion of cells per scRNAseq sample in the *mitfa⁺aox5*ʰⁱ cycling subpopulations (*mitfa⁺aox5*ʰⁱ S phase and *mitfa⁺aox5*ʰⁱ G2/M subpopulations combined) during regeneration (Day 0 = 7989, Day 1 = 4004, Day 2 = 4510, Day 3 = 5224, Day 5 = 3483, Day 10 = 4243 total cells per sample). p values calculated using differential proportion analysis (*Farbehi et al., 2019*), **p < 0.01; ****p < 0.0001; ns, not significant. (**E**) Dot plot of pigment cell and cell cycle marker genes differentially expressed across S adj, cycling, and G2/M adj *mitfa⁺aox5*ʰⁱ subpopulations during melanocyte regeneration. Dot sizes represent percentage of cells in the cluster expressing the marker and coloring represents average expression.

The online version of this article includes the following source data and figure supplement(s) for figure 6:

**Source data 1.** Proportion of cells per scRNAseq sample in the *mitfa⁺aox5*ʰⁱ cycling subpopulations during regeneration.

**Figure supplement 1.** Gene expression in *mitfa⁺aox5*ʰⁱ cycling subpopulations.

that MP-0 is a more truncal, stem-like subpopulation that can also backfill the MP-1 subpopulation during regeneration. The benefit to having two subpopulations of progenitors is unclear, although it appears to enable regeneration of new melanocytes by two different routes, the differential regulation of which may be important for regeneration to occur proficiently under different circumstances. Regardless, the resolution afforded by scRNAseq indicates that the MP-0 and MP-1 subpopulations are present in unperturbed animals and primed to adopt different fates when activated in response to melanocyte injury.

Heterogeneity may also be evident by the additional *mitfa⁺aox5*ʰⁱ G2/M adj subpopulation that likely arises via cell divisions during regeneration. There are reasons to think that this could be a progenitor subpopulation. Firstly, these cells arose in response to specific ablation of melanocytes. Secondly, this subpopulation expresses markers that are associated with multipotent pigment progenitors cells found during development (*Brombin et al., 2022*; *Brunsdon et al., 2022*; *Budi et al., 2011*; *Johansson et al., 2020*; *Saunders et al., 2019*; *Subkhankulova et al., 2023*). Thirdly, although this subpopulation expresses *aox5* and some other markers associated with xanthophores, we showed that differentiated xanthophores are not ablated by the melanocyte-ablating drug neocuproine and this *mitfa⁺aox5*ʰⁱ subpopulation does not make new pigmented xanthophores following neocuproine treatment. However, current observations cannot definitively determine the potency and fates adopted by these cells. One possibility is that these cells are indeed progenitors that arise through cell divisions, are in an as yet undefined way lineally related to MP-0 and MP-1 subpopulations, and ultimately give rise to new melanocytes during additional rounds of regeneration. Given their expression of markers associated with multipotent pigment cell progenitors, these cells could be multipotent but fated toward the melanocyte lineage following melanocyte-specific ablation. However, we cannot exclude the possibility that these cells are another cell type. For example, there is a type of partially differentiated xanthophores that populate adult melanocyte stripes (*McMenamin et al., 2014*). At least some of these cells arise from embryonic xanthophores that transitioned through a cryptic and proliferative state (*McMenamin et al., 2014*). That the descendants remain partially differentiated could indicate that they are in more of a xanthoblast state and maintain proliferative capacity (*Eom et al., 2015*). It is possible that some or all of the cells in question are melanocyte stripe-resident, partially differentiated xanthophores that arise: (1) from cell divisions that are triggered by loss of interactions with melanocytes, or (2) simply to fill space that is vacated due to melanocyte death. Such causes for partially differentiated xanthophore divisions have not been documented, but nonetheless this possibility must be considered given the *mitfa* and *aox5* expression and proliferative potential of these cells. Transcriptional profiles of 'cryptic' xanthophores are not available to help clarify the nature of these cells. Lastly, the relationship between adult progenitor populations – MP-0, MP-1 and, potentially, *mitfa⁺aox5*ʰⁱ G2/M adj – and other progenitors present at earlier developmental stages is unclear and could be defined through additional long-term lineage tracing studies. In particular, previous examinations of pigment cell progenitors in developing zebrafish have identified dorsal root ganglion-associated pigment cell progenitors in larvae that contribute to adult pigmentation patterns (*Budi et al., 2011*; *Dooley et al., 2013*; *Singh et al., 2016*). It is possible that these cells give rise to the adult progenitors we have identified. The further alignment of cell types that have been observed in vivo and cell subpopulations defined through expression profiling is a necessary route for understanding the complex relationship between stem and progenitor cells in development, homeostasis, and regeneration.

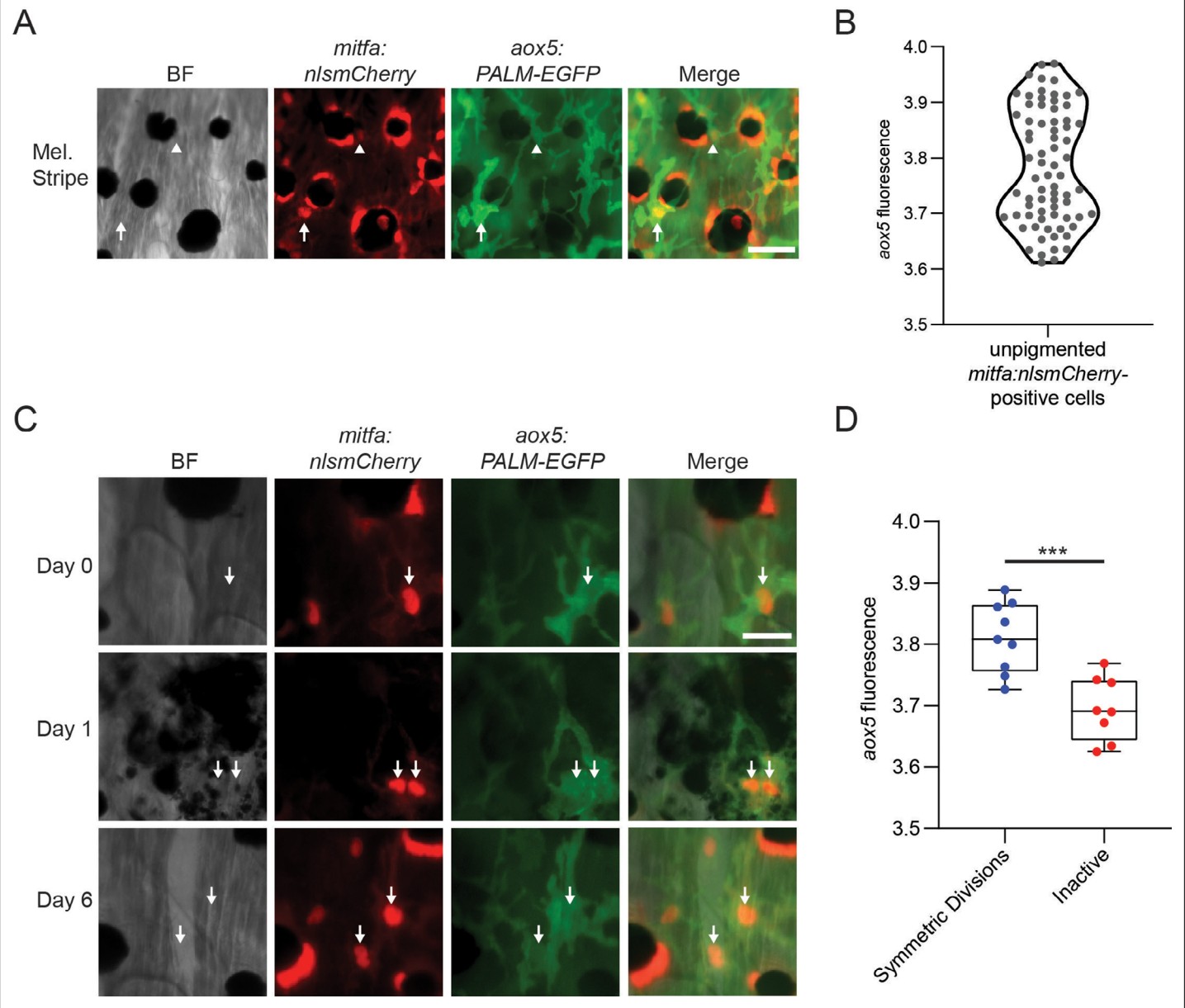

**Figure 7.** *aox5* expression predicts in vivo progenitor cell fate. (**A**) Representative image of progenitors expressing different levels of *aox5* promoter-driven PALM-EGFP. *mitfa⁺aox5ʰⁱ* (arrowhead), *mitfa⁺aox5ˡᵒ* (arrow). Scale bar = 50 µm. (**B**) Quantification of PALM-EGFP fluorescence intensity indicates groups of progenitors that express lower and higher levels of *aox5*. Mean pixel intensity per area; intensity values log normalized. *n* = 73 cells from 5 animals. (**C**) Images of an *mitfa⁺aox5ʰⁱ* cell that underwent mitosis following melanocyte destruction. Scale bar = 30 µm. (**D**) Comparison of *aox5* intensity in cells that underwent symmetric divisions or remained inactive. Mean pixel intensity per area; intensity values log normalized. *n* = 17 cells from 5 animals. p values calculated by Student's *t*-test, ***p < 0.001.

The online version of this article includes the following source data and figure supplement(s) for figure 7:

**Source data 1.** *aox5* expression *in* progenitors.

**Figure supplement 1.** Lack of interstripe xanthophore divisions following neocuproine treatment.

Our investigation of KIT signaling sheds light on how the process of melanocyte regeneration is controlled. KIT signaling has well-known roles in melanocyte development in promoting the differentiation, survival and migration of melanocytes (*Rawls and Johnson, 2003*; *Yoshida et al., 2001*). Here, we have found that KIT signaling is reactivated during melanocyte regeneration to enable progenitors to generate new melanocytes. Specifically, shortly after melanocyte ablation we observed increased expression of the *kitlga* ligand, and the increased expression of *kitlga* was coupled with

an increase in MAPK signaling in progenitors. KIT receptor mutants showed a profound reduction in cells in the direct differentiation subpopulation and a less marked, but nonetheless significant, reduction in the cycling differentiation subpopulation. In these mutants there were no changes in the *mitfa⁺aox5ʰⁱ* cycling subpopulation. For some context, studies of Kit dependence during the hair cycle in the mouse found that McSCs could be divided into two groups, a Kit-independent quiescent McSC pool and an activated, cycling, Kit-dependent McSC pool that is derived from the quiescent McSCs (*Ueno et al., 2014*). In this study, the quiescent pool of McSCs was spared during treatment with anti-Kit neutralizing antibody whereas the activated, cycling McSCs died as a result of Kit inhibition. In our study we find that MP-0 and MP-1 subpopulations are similarly unaffected by *kita* loss of function. However, the subpopulations that are products of MP-0 and MP-1 activation during regeneration – direct differentiation and cycling differentiation subpopulations – are both impacted by *kita* loss. Thus, whereas quiescent McSCs in the mouse and melanocyte progenitors in the zebrafish are insensitive to loss of Kit activity, cells derived from these subpopulations become sensitive to loss of Kit. Interestingly, although melanocyte progenitors in our study were unaffected by loss of *kita*, we nonetheless found that MAPK signaling was decreased in progenitors as a result of *kita* loss. This suggests that, while zebrafish subpopulations of progenitors might upregulate KIT-dependent MAPK signaling during regeneration, it is the products of these progenitors that are deficient. This indicates a potential role for Kit in the specification or survival of cycling differentiation and direct differentiation subpopulations during regeneration.

Overall, our study has found unexpected heterogeneity of progenitor subpopulations, defined new intermediate subpopulations involved in regeneration, and discovered cellular trajectories that link these subpopulations during the regeneration process. Along with providing a greater understanding of melanocyte regeneration in zebrafish and tissue regeneration in general, these findings are relevant to pigmentary disorders and melanoma. In recovery from vitiligo, after the immune-mediated destruction of melanocytes, melanocyte stem cells must be activated and produce differentiated melanocytes that migrate from the hair follicle niche into interfollicular epidermis (*Eby et al., 2014*; *Nishimura et al., 2005*; *Nishimura et al., 2002*; *Nishimura et al., 2010*). By illuminating cellular transitions and molecular signaling pathways involved in regeneration, our studies could assist in the development of treatments or therapeutics for pigmentary disorders to encourage melanocyte progenitor activation, migration, and differentiation. In melanoma, stem cell-like identities are implicated in tumor maintenance and drug resistance. In single-cell analysis of human melanomas, a subpopulation of cells expressing high levels of AXL, a putative cancer stem cell marker, and low levels of MITF were present in treatment-naive human melanoma samples (*Tirosh et al., 2016*). These AXL-high MITF-low cells were admixed with AXL-low MITF-high differentiated cells, demonstrating cellular heterogeneity present in untreated melanomas. Furthermore, samples from patients after relapse were enriched for this AXL-high state, furthering the concept that stem-like cells facilitate relapse. Similarly, zebrafish melanoma models have demonstrated the role of a stem-like minimal residual disease cells in facilitating relapse (*Travnickova et al., 2019*). In this study, MITF-low zebrafish melanoma cells had striking transcriptional similarities to invasive melanoma signatures found in humans following treatment with small molecule inhibitors of MEK or BRAF (*Travnickova et al., 2019*). Additional studies have found stem-like properties in melanoma cells involved in therapeutic resistance and tumor relapse (*Boshuizen et al., 2020*; *Konieczkowski et al., 2014*; *Mehta et al., 2018*; *Rambow et al., 2018*; *Sharma et al., 2017*). Our study provides a reference to which melanoma cells, as well as cells involved in pigmentary diseases such as vitiligo, can be compared to for identifying relationships between and conserved pathways shared by melanocyte progenitors and cells involved in melanocytic diseases.

## Methods

**Key resources table**

| Reagent type (species) or resource | Designation | Source or reference | Identifiers | Additional information |
|---|---|---|---|---|
| Chemical compound, drug | Neocuproine | Sigma-Aldrich | ID_source:N1501 | |

*Continued on next page*

*Continued*

| Reagent type (species) or resource | Designation | Source or reference | Identifiers | Additional information |
|---|---|---|---|---|
| Chemical compound, drug | (−)-Epinephrine (+)-bitartrate salt, 98+% | Acros Organics | ID_source:AC430140010 | |
| Chemical compound, drug | Tricaine methanesulfonate | Syndel | | |
| Chemical compound, drug | TM Liberase | Sigma-Aldrich | ID_source:LIBTM-RO | |
| Strain (*Danio rerio*) | WT (AB) | Gift. | | |
| Strain (*Danio rerio*) | *kita(lf)* | Gift. *Parichy et al., 1999* PMID:10393121 | | |
| Strain (*Danio rerio*) | *kitlga(lf)* | Gift. *Hultman et al., 2007* PMID:17257055 | | |
| Strain (*Danio rerio*) | *Tg(mitfa:nlsEGFP)* | This paper | | *Figure 1*, Methods: DNA constructs, Transgenic Fish |
| Strain (*Danio rerio*) | *Tg(mitfa:nlsmCherry)* | This paper | | *Figure 1*, Methods: DNA constructs, Transgenic Fish |
| Strain (*Danio rerio*) | *Tg(mitfa:ERKKTR-mClover)* | This paper, *Mayr et al., 2018* PMID:30320107 | | *Figure 5*, Methods: DNA constructs, Transgenic Fish |
| Strain (*Danio rerio*) | *Tg(aox5:PALM-EGFP)* | Gift. *Eom et al., 2015* PMID:26701906 | | |
| Gene (*Danio rerio*) | *kitlga* | *Parichy et al., 1999* PMID:10393121 | | Amplified from cDNA |
| Gene (*Danio rerio*) | *kita* | *Hultman et al., 2007* PMID:17257055 | | Amplified from cDNA |
| Commercial kit | SYBR Green | Thermo Fisher | ID_source:4472908 | |
| Commercial kit | SuperScript III First-Strand Synthesis System for RT-PCR | Invitrogen | ID_source:18080-051 | |
| Commercial kit | Chromium Next GEM Single Cell 3' GEM, Library, and Gel Bead Kit v3.1 | 10× Genomics | ID_source:1000128 | |
| Software, algorithm | Cell Ranger | 10× Genomics | V3.0.0 | |
| Software, algorithm | Seurat | *Stuart et al., 2019* PMID:31178118 | V3 | https://github.com/satijalab/seurat; *satijalab, 2022* |
| Software, algorithm | Monocle3 | *Saunders et al., 2019* PMID:31140974 | V3 | https://github.com/cole-trapnell-lab/monocle3; *Trapnell et al., 2022* |
| Software, algorithm | SCANPY | *La Manno et al., 2018* PMID:30089906 | V0.2.0 | https://github.com/theislab/scvelo; *Theis Lab, 2017* |
| Software, algorithm | scvelo | *Bergen et al., 2020* PMID:32747759 | V1.6.0 | https://github.com/theislab/scanpy; *scverse, 2022* |
| Software, algorithm | NicheNetR | *Browaeys et al., 2020* PMID:31819264 | | https://github.com/saeyslab/nichenetr; *DaMBi, 2023* |
| Software, algorithm | cellphonedb | *Garcia-Alonso et al., 2021* PMID:34857954 | | https://github.com/Teichlab/cellphonedb; *Teichmann Group, 2021* |
| Other | GEO data: scRNAseq of human skin | *Joost et al., 2020* PMID:23109378 | | GSE129218 |
| Other | GEO data: scRNAseq of zebrafish skin | *Saunders et al., 2019* PMID:31140974 | | GSE131136 |
| Other | GEO data: scRNAseq of murine skin | *Infarinato et al., 2020* PMID:33184221 | | GSE147299 |

## Fish stocks and husbandry

Fish stocks were maintained at 28.5°C on a 14L:10D light cycle (*Westerfield, 2007*). The strains used in this study were AB, referred to as wild-type or WT, *kita(b5)*, referred to as *kita(lf)* (*Parichy et al., 1999*), and *kitlga(tc244b)*, referred to as *kitlga(lf)* (*Hultman et al., 2007*). Previously published strains used include: *Tg(aox5:PALM-eGFP)* (*Eom et al., 2015*) and *Tg(mitfa:BRAFV600E)* (*Ceol et al., 2011*). Construction of new strains generated for this study is detailed below.

## DNA constructs

DNA constructs were built using the Gateway system (Life Technologies). Previously published entry clones used in this research are: pENTRP4P1r-Pmitfa (*Ceol et al., 2011*), pENTR-ERKKTRClover (*Mayr et al., 2018*), and p3E-polyA (*Kwan et al., 2007*). Previously published destination vectors include: pDestTol2pA2 (*Kwan et al., 2007*). Using the entry clones described above, the following constructs were built with standard multisite Gateway reactions: pDestTol2pA2-Pmitfa:ERKKTRClover:pA, pDestTol2pA2-Pmitfa:nlsEGFP:pA, and pDestTol2pA2-Pmitfa:nlsmCherry:pA.

## Microinjection and transgenic fish

For transposon-mediated integration, 25 pg of a construct was injected with 25 pg of Tol2 transposase mRNA into zebrafish embryos at the single-cell stage. For injection into an existing Tol2-generated line, constructs were linearized and injected into single-cell embryos without transposase. To create the *Tg(mitfa:nlsEGFP)* transgenic line, pDestTol2pA2-Pmitfa:nlsEGFP:pA was injected into wild-type embryos, EGFP-positive larvae selected and the resulting adults outcrossed to wild-type animals. EGFP-positive larvae from these crosses were further outcrossed to wild-type animals to establish a stable *Tg(mitfa:nlsEGFP)* transgenic line.

## Melanocyte destruction and single-cell serial imaging analysis

Adult zebrafish were treated with 750 nM neocuproine in a beaker for 24 hr and then kept in individual tanks filled with fish water after drug washout. Prior to imaging fish were treated with 1 mg/ml (−)-epinephrine (+)-bitartrate salt (Acros Oragnics) for 2 min and then anesthetized with 0.17 mg/ml tricaine. Fish were then placed on their sides in a plastic Petri dish and imaged in the same locations over time using anatomical and cellular landmarks. Fish were viewed with a Leica DM550B microscope and images were captured with a Leica DFC365FX camera. Representative examples of direct

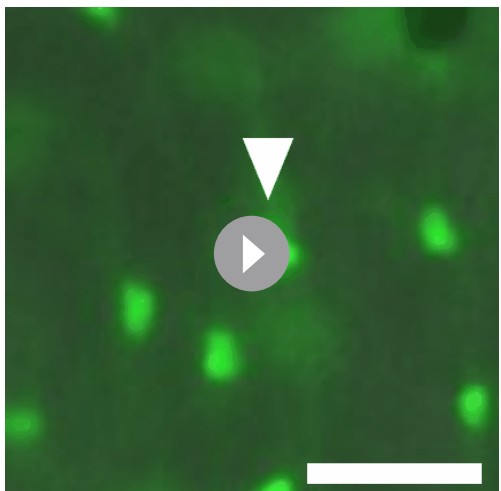

**Animation 1.** Serial imaging of direct differentiation. Representative images from serial imaging of an unpigmented *mitfa*-expressing cell undergoing direct differentiation in the melanocyte stripe of a *Tg(mitfa:nlsEGFP)* zebrafish. Merged brightfield and GFP. White arrowheads indicate traced cell. Scale bar = 50 μm.

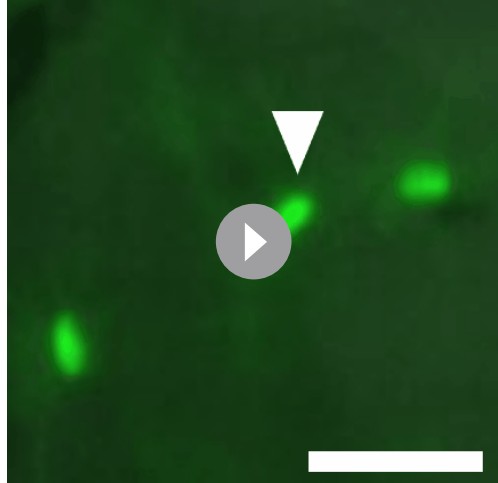

**Animation 2.** Serial imaging of symmetric division. Representative images from serial imaging of an unpigmented *mitfa*-expressing cell undergoing symmetric division in the melanocyte stripe of a *Tg(mitfa:nlsEGFP)* zebrafish. Merged brightfield and GFP. White arrowheads indicate traced cell and subsequent daughter cells. Scale bar = 50 μm.

differentiation (Animation 1) and symmetric division (Animation 2) fates are provided. The fish were then placed in fish water for recovery.

## Imaging and quantitative analysis

Brightfield images were adjusted for color balance, contrast, and brightness for clarity. Cells were counted along the flank's middle melanocyte stripe (second stripe down from the dorsum) under ×20 magnification using FIJI ImageJ. Fractional regeneration was calculated as a ratio of the number of melanocytes within the stripe region at the specified time point relative to the same region prior to melanocyte destruction via neocuproine. Student's *t*-tests were performed using GraphPad Prism 9. Serial imaging was performed by aligning daily brightfield and fluorescent images using anatomical and cellular landmarks. *mitfa:nlsEGFP*-expressing cells were traced from day 0 (prior to melanocyte ablation) until day 10. Differentiation was recognized by melanization, and self-renewal was recognized by a division in which neither daughter cell differentiated.

## RT-PCR

Adult WT fish were dissected, skin tissue was placed in TRIzol (Thermo Fisher) and mechanically dissociated before RNA isolation and purification using the RNeasy kit (QIAGEN) per manufacturer's protocol. Full-length cDNA was synthesized with SuperScript III First Strand Synthesis kit (Thermo Fisher). Reaction mixes were comprised of SYBR Green RT-PCR master mix (Thermo Fisher), primers (*Supplementary file 3*), and 25 ng cDNA. Analysis was performed using a StepOnePlus Real Time PCR System (Applied Biosystems). Samples were normalized to *β-actin* for loading control and *zfk8* for a surgical sampling control. Fold changes were calculated using ΔΔCt in Microsoft Excel.

## ERKKTR

To quantify ERK signaling, mClover intensity was measured in the nucleus and cytoplasm using Fiji's measure intensity tool. Nuclear localization was defined by nlsmCherry expression. The ratio of cytoplasmic to nuclear signal intensity was calculated using Microsoft Excel. Progenitors were randomly selected and quantified at time points before, during, and after melanocyte regeneration.

## *aox5* intensity of progenitor fates

Progenitors were traced during regeneration and assayed for cell fate. Differentiation was recognized by melanization, and self-renewal was recognized by a division in which neither daughter cell differentiated. The *aox5* fluorescence intensity of these cells was then measured using Fiji. *aox5:PALM-eGFP* intensity was calculated in ImageJ by determining the mean intensity within a cell, subtracting the background, and log transforming the signal. Differences in *aox5* intensity between traced fates were visualized in Prism 9.

## scRNAseq sample prep and FACS enrichment

Prior to zebrafish dissection the animals were euthanized with 0.85 mg/ml tricaine. Scales were then removed from the lateral aspects of the animal with forceps, and a scalpel was used to cut the epidermal and dermal tissue from the underlying muscle and adipose tissue. Skin was then peeled away with forceps. The skin retrieved was caudal to the gills, dorsal to the lower melanocyte stripe, ventral to the top melanocyte stripe, and rostral to the anal fin. Dissected skin was enzymatically dissociated with Liberase TM (Sigma-Aldrich LIBTM-RO, 0.25 mg/ml in Dulbecco's phosphate-buffered saline) at 32°C for 20 min followed by manual trituration with a glass pipette for 2 min. Cell suspensions were then filtered through a 70-μm Nylon cell strainer. This single-cell suspension was then spun down for 5 min at 1500 pm in an Eppendorf 5810 R swing bucket centrifuge and resuspended in 0.1% bovine serum albumin (BSA)/5% fetal bovine serum (FBS) in dPBS for FACS enrichment. Singlets were then isolated using a series of FSC-A versus SSC-A and FSC-A versus FSC-H gates. We then enriched for our *mitfa:nlsEGFP*-expressing cells using unlabeled WT animals as a negative control. All samples were kept on ice in 0.1% BSA/5% FBS except during enzymatic dissociation. All surfaces encountering the cell solution were coated in 1% BSA before use.

## scRNAseq regeneration sampling, library construction sequencing

For each time point before (day 0) and during (days 1, 2, 3, 5, and 10) regeneration we captured transcriptomes from WT and *kita(lf)* fish. To generate sufficient material for single-cell capture 30 zebrafish

lateral skins were dissected for each time point and an estimated 10,000 EGFP-positive cells per sample were obtained via FACS. For each sample, we targeted 4000–6000 cells for capture using the 10× Genomics Chromium platform with one sample per lane. Libraries were prepared using the Single Cell 3′ kit (v3.1). Quality control and quantification assays were performed using a Qubit fluorometer (Thermo Fisher) and a fragment analyzer (Agilent). Libraries were sequenced on an Illumina NextSeq 500 using 75-cycle, high output kits with read 1: 28 cycles, index: 8 cycles, read 2: 50 cycles. Each sample was sequenced to an average depth of 216 million total reads resulting in an average read depth of approximately 59,000 reads/cell.

## scRNAseq sequencing analysis

We ran the below pipelines in the DolphinNext environment (https://github.com/UMMS-Biocore/dolphinnext; *Yukselen and Kucukural, 2022*) using the Massachusetts Green high Performance Computing Cluster (*Yukselen et al., 2020*). Raw base call (BCL) files were analyzed using CellRanger (version 3.1.0). Cell Ranger 'mkfastq' was used to generate FASTQ files and 'count' was used to generate raw feature-barcode matrices aligned to the Lawson Lab zebrafish transcriptome annotation v4.3 (*Lawson et al., 2020*). Cell Ranger defaults for selecting cell-associated barcodes versus background empty droplet barcodes were used.

## scRNAseq downstream data analysis

Filtering retained cells that expressed between 200 and 6000 unique genes, and a mitochondrial fraction less than 10%. These parameters resulted in retention of 29,453 WT cells and 24,724 *kita(lf)* cells across 14 total samples. Each dataset was further pre-processed by log normalizing each dataset with a scale factor of 10,000 and finding 2,000 variable features using the default 'vst' method in seruat3.0 (*Stuart et al., 2019*). Datasets were integrated using the WT datasets as references in the 'FindAnchors' step (*Stuart et al., 2019*). After integration the integrated dataset was scaled to regress out differences driven by inequalities in total RNA count and mitochondrial capture. The top 60 principal components were used for construction of a UMAP and neighbor finding. During clustering we utilized a resolution parameter of 1.0. The resulting 33 Louvain clusters were visualized in 2D and/or 3D space and were annotated using known biological cell-type markers. Visualization of the UMAP and calculation of DEGs split by sample revealed no observable strong batch effects. Changing any of the above parameters yielded similar cell-type identifications and clustering structures. Both 2000 and 4000 variable features revealed similar results in final cell clustering and DEG analysis. Similarly, using between 20 and 80 principal components and a resolution parameter between 0.5 and 1.2 revealed similar UMAP visualizations and clustering, underlining the robustness of the approach.

## NicheNetR

Our analysis identifying potential ligand/receptor pairs regulating progenitor behavior during regeneration utilizes the open-source R implementation of NicheNetR available at GitHub (github.com/saeyslab/nichenetr). We followed the default parameters for average logFC and percent expression. For elucidating ligand/receptor interactions we assigned all non-progenitor cell populations as defined by clustering in Seurat as potential 'sender cells' and progenitors as 'receiver cells'. We then filtered down receptor ligand pairs to the 'bona fide' literature-supported interactions using NicheNetR's built in lists.

## Trajectory analysis and pseudotime

Graph trajectory learning as well as pseudotime was calculated using default parameters on a UMAP visualization of progenitors and melanocytes obtained in Seurat and imported into monocle3 (*Saunders, 2019*). To normalize for differences in real-time sampling cells were randomly downsampled to the lowest common denominator. This allowed that real-time sample size differences would not impact pseudotime distribution calculation. RNA velocity splicing mechanics were calculated using default parameters in velocyto (*La Manno et al., 2018*). Then velocity embeddings and streams were calculated on the Seurat derived UMAP embedding using default parameters in scvelo (*Bergen et al., 2020*).

## Integration of scRNAseq data with existing datasets

Data from *sox-10*-positive cells (*Saunders et al., 2019*) were obtained from the Gene Expression Omnibus (GSE131136). Using metadata uploaded by the authors, hypothyroid samples were removed

and euthyroid samples from multiple time points were filtered to cells expressing between 200 and 6000 genes and a mitochondrial fraction less than 10%. Unlike the approach listed above with *kita(lf)* and WT samples, these euthyroid samples were then mapped onto the UMAP and clusters calculated as part of *Figure 1C* using the Seurat MapQuery wrapper (*Hao et al., 2021*). In this the TransferData function was used to classify cells from *Saunders et al., 2019* based on the WT cells obtained in *Figure 1C* of this paper. These predicted cell types were then added to the *Saunders et al., 2019* euthyroid cells as 'predicted cell-types'. Lastly, we computed a reference UMAP model using the UMAP present in *Figure 1* and projected the *Saunders et al., 2019* data, complete with predicted cell-type classifications, onto this UMAP. Feature plots of known cell markers demonstrated the high correlation in gene expression.

### Marker calculation and comparison to murine datasets
Murine sample data were obtained from GEO GSE147299. Data were processed in Seurat using methods described above. Cells from *BMP*-KO samples as well as non-melanocyte lineage keratinocytes were then removed. Signature expression was calculated using 'FindMarkers' in each dataset. Markers were then converted to murine orthologs using HGNC orthology tables as a key. Zebrafish signature scores were computed for murine data using Seurat's 'AddModuleScore' using the top 100 differentially expressed marker genes from each zebrafish cluster. Scores between cell types were visualized using a DotPlot.

### Cell cycle scoring
Cell cycle scores were calculated using Seurat's 'CellCycleScoring' module. The 'AddModule' function within this module was used to calculate scores for S and G2M phases using known marker genes (such as *pcna* for S phase and *cdk1* for G2M phase) (*Tirosh et al., 2016*). If a cell was enriched for S or G2M module genes (S.Score or G2M.Score >0), it was designated as whichever phase scored higher. If the cell was not enriched for S or G2M phase genes (S.Score and G2M.Score <0), it was designated as a G1 cell (*Stuart et al., 2019*).

### Comparisons to fibroblast and keratinocyte data
Murine sample data of fibroblasts and keratinocytes were obtained from GEO GSE129218. Data were processed in Seurat using methods described above. Gene expression profiles for canonical fibroblast and keratinocyte marker genes were visualized alongside *Kitlg* expression using feature plots.

### Statistics
Student's *t*-tests were performed using GraphPad Prism 9. Wilcoxon rank-sum tests were calculated in R[version 4.0.0] (*R Development Core Team, 2020*). Cluster population shifts analyzed using differential proportion analysis using default parameters (*Farbehi et al., 2019*).

---

## Additional information

### Funding

| Funder | Grant reference number | Author |
|---|---|---|
| National Institute of Arthritis and Musculoskeletal and Skin Diseases | R01 AR081355 | William Tyler Frantz |
| National Institute of General Medical Sciences | T32 GM107000 | William Tyler Frantz |
| National Cancer Institute | T32 CA130807 | William Tyler Frantz |

The funders had no role in study design, data collection, and interpretation, or the decision to submit the work for publication.

## Author contributions

William Tyler Frantz, Conceptualization, Data curation, Formal analysis, Validation, Investigation, Visualization, Methodology, Writing – original draft, Writing – review and editing; Sharanya Iyengar, James Neiswender, Investigation, Methodology; Alyssa Cousineau, Data curation, Investigation, Methodology; René Maehr, Supervision, Methodology; Craig J Ceol, Conceptualization, Resources, Supervision, Funding acquisition, Investigation, Methodology, Writing – original draft, Project administration, Writing – review and editing

## Author ORCIDs

William Tyler Frantz ⓘ http://orcid.org/0000-0003-1207-9652
René Maehr ⓘ http://orcid.org/0000-0002-9520-3382
Craig J Ceol ⓘ http://orcid.org/0000-0002-7188-7580

## Ethics

This study was performed in strict accordance with the recommendations in the Guide for the Care and Use of Laboratory Animals of the National Institutes of Health. Zebrafish were handled in accordance with protocols approved by the University of Massachusetts Medical School IACUC protocol (A-2171). For procedures, including imaging and genotyping, animals were anesthetized in 0.17% tricaine or euthanized by overdose of tricaine. Every effort was made to minimize suffering.

## Decision letter and Author response

Decision letter https://doi.org/10.7554/eLife.78942.sa1
Author response https://doi.org/10.7554/eLife.78942.sa2

---

# Additional files

## Supplementary files

• Supplementary file 1. Table of single-cell RNA-sequencing cluster markers and sample statistics. Cluster labels (Table 1, Tab 1) and top 100 differentially expressed marker genes (Table 1, Tab 2) from *Figure 1C*, and number of cells sampled from each condition (Table 1, Tab 3) from *Figure 1A, B*.

• Supplementary file 2. Table of cluster-specific differentially expressed genes. Top 100 differentially expressed marker genes from melanocyte progenitor-0 (MP-0), melanocyte progenitor-1 (MP-1), cycling differentiation, direct differentiation, and melanocyte subclusters found in *Figure 2A* which were used to calculate enrichment scores on murine data in *Figure 3— figure supplement 1*.

• Supplementary file 3. Table of RT-PCR primer sequences. Previously validated primer sequences for *kitlga* and *kita* utilized in *Figure 3* (*Hultman et al., 2007*; *Parichy et al., 1999*).

• MDAR checklist

## Data availability

Sequencing data have been deposited in GEO under accession code: GSE190115. Other Source Data files have been provided for individual figures.

The following dataset was generated:

| Author(s) | Year | Dataset title | Dataset URL | Database and Identifier |
|---|---|---|---|---|
| Frantz WT, Iyengar S, Neiswender J, Cousineau A, Maehr R, Ceol CJ | 2022 | Dissection of melanocyte stem cell transcriptomes during melanocyte regeneration in adult zebrafish | https://www.ncbi.nlm.nih.gov/geo/query/acc.cgi?&acc=GSE190115 | NCBI Gene Expression Omnibus, GSE190115 |

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
