## [Editor Report]

This valuable study advances our understanding of heterogeneous transcriptomic states and genetic requirements of skin-resident pigment cells and pigment cell progenitors in adult zebrafish, relevant to regenerative biology and melanoma origins. The single-cell and bioinformatic analyses and the use of mutants and regeneration assays are carefully done and appropriately interpreted. The work provides useful new observations that will be of interest to researchers focused on the basic biology of adult pigmentary phenotypes and their homeostasis, as well as those pursuing translational aspects of regeneration and melanoma origins and treatments.

---

## [Decision Letter]

**Decision letter after peer review:**

Thank you for submitting your article "Stem cell heterogeneity and reiteration of developmental signaling underlie melanocyte regeneration in zebrafish" for consideration by *eLife*. Your article has been reviewed by 3 peer reviewers, and the evaluation has been overseen by a Reviewing Editor and Didier Stainier as the Senior Editor. The following individuals involved in the review of your submission have agreed to reveal their identity: Melissa L. Harris (Reviewer #3).

The consensus opinion is that the work is interesting, makes a valuable contribution to the field, and is appropriate for publication in *eLife*. The reviews are thorough and identify several issues needing clarification, minor modifications, additional citations, etc., all of which are reasonable and all of which will improve the paper in its final version.

The one major sticking point that is essential to address concerns the identity of mitfa+ aox5-hi cells as the reviewers (and I) are not convinced these are bona fide McSC as opposed to "cryptic" xanthophores. Indeed one might expect cryptic xanthophores to have the transcriptomic profile shown here, clustering with other xanthophores, and to proliferate after melanocyte ablation simply because there is space to do so. The ambiguity could be addressed by tempering relevant portions of the text OR by providing experimental evidence that shows the differentiation of these cells into melanophores (which would be exciting, if true).

*Reviewer #2 (Recommendations for the authors):*

1) Identity/Xanthophores: The authors list 14 clusters that identify as xanthophores and call them mitfa+aox5+ cells on the UMAP (Figure 1). Later the authors follow mitfa+aox5+ cells dividing by scRNA-seq, and in the stripes and interpret these as McSCs (Figure 6, Figure 7). It seems likely these are simply dividing xanthophores? More evidence is required to link these cells to melanocytes, and melanocyte stem cells.

2) scRNA-seq analysis: Please explain some of the choices taken in the scRNA-seq pipeline.

a. Cells with 200 expressed genes are considered high quality. Can the authors justify this figure?

b. We wonder whether the integration between WT and Kita datasets missed any other major differences between WT and mutant cells? Were the datasets so similar even before the integration?

3) Statistics: Although the authors used appropriate statistical tools in the analysis of the scRNA-seq datasets, we are concerned about the stats used in the plots generated with GraphPad Prism. The Student's t-test can be used when comparing the mean of two datasets with normal distribution. However, it cannot be used for multiple comparisons. Please review the choice of their statistical test for Figures 3C, 4A, 4E, 5B, 5D, 5F, and S5B, and indicate the missing statistical test in Figures S1C, S2D, S6B.

[Please use ANOVA or similar for initial analyses to assess whether significant differences are present overall, and appropriate post hoc comparisons, like Tukey-Kramer, if warranted. Include values of test statistics (e.g., F) with degrees of freedom for overall tests. -Parichy]

4) Context in the field:

a. The authors directly examine adult skin, which has not been done in depth before, and provide an important dataset resource. However, it is important to interpret the data within the context of the wider melanocyte stem cell field (i.e. DRG-associated and daughter cells lining the peripheral nerves as shown by Budi et al., 2011; Singh et al., 2014; Singh et al., 2016, Brombin et al., 2022). These cells are also kit-dependent (Dooley et al., 2013). It is important to address how the melanocyte adult skin progenitors relate to the DRG-associated progenitors. One idea is that the cells that line nerves are a source of melanocytes. Not all the pigment progenitors described by Saunders and colleagues are skin-associated. Could these progenitors contribute to the regeneration in adult fish as previously shown (Budi et al., 2011)?

b. We have concerns with the use of the term "stem cell", and believe the data is more in line with "Progenitor" as they have previously called these cells in Iyengar et al., 2015.

c. For their cycling differentiation cells, it might be appropriate to cite other work showing division of differentiating melanocytes as well (e.g. Taylor et al. 2011)

5) Melanoma: As the authors rightfully claim, the study of McSC might be beneficial to the understanding of how these contribute to melanoma, however, they do not make an actual link to melanoma signatures. Perhaps they could cite and discuss some of the papers that show McSCs and progenitors in zebrafish are relevant to melanoma?

*Reviewer #3 (Recommendations for the authors):*

This study uses a nice combination of -omics and zebrafish genetics/lineage tracing to heighten our understanding of McSCs and their potential. The manuscript extends our understanding of existing mechanisms (e.g., KIT signaling in McSCs) but also makes small leaps to novel discovery (e.g., identification and function of axo5 self-renewing McSCs). The extensive single-cell datasets generated by this study will be of interest to researchers interested in pigment regeneration, stem cell-based therapeutics for pigment disorders, and the basic biology of stem cells and their heterogeneity. Overall, the data presented seems to support the claims.

---

## [Author Response]

The reviewers have discussed their reviews with one another, and the Reviewing Editor has drafted this to help you prepare a revised submission.The consensus opinion is that the work is interesting, makes a valuable contribution to the field, and is appropriate for publication in eLife. The reviews are thorough and identify several issues needing clarification, minor modifications, additional citations, etc., all of which are reasonable and all of which will improve the paper in its final version.The one major sticking point that is essential to address concerns the identity of mitfa+ aox5-hi cells as the reviewers (and I) are not convinced these are bona fide McSC as opposed to "cryptic" xanthophores. Indeed one might expect cryptic xanthophores to have the transcriptomic profile shown here, clustering with other xanthophores, and to proliferate after melanocyte ablation simply because there is space to do so. The ambiguity could be addressed by tempering relevant portions of the text OR by providing experimental evidence that shows the differentiation of these cells into melanophores (which would be exciting, if true).

We agree with the editor and reviewers that the identities of the mitfa^+^aox5^hi^ cells and the interplay between these cells and the mitfa^+^aox5^lo^ cells is a fascinating, and originally unexpected, aspect of this manuscript. The issue, as we see it, is whether mitfa+aox5hi cells that arise via cell division during regeneration are multipotent pigment cell progenitors or ‘cryptic’ xanthophores. The experiments we have performed to address this ambiguity have not worked for technical reasons, so we have tempered text in the relevant Results and Discussion sections to leave both options open. We have backed off from calling these cells progenitors but have included additional data showing that they (i.e. the mitfa^+^aox5^hi^ subpopulation of cells that we believe are daughters of mitfa+aox5hi cycling cells) express multiple markers associated with multipotent pigment cell progenitors that have been characterized in developing zebrafish. Our expanded Discussion is as follows:

“Heterogeneity may also be evident by the additional mitfa^+^aox5^hi^ G2/M adj subpopulation that likely arises via cell divisions during regeneration. There are reasons to think that this could be a progenitor subpopulation. Firstly, these cells arose in response to specific ablation of melanocytes. Secondly, this subpopulation expresses markers that are associated with multipotent pigment progenitors cells found during development (Budi, et al., 2011; Saunders, et al., 2019). Thirdly, although this subpopulation expresses aox5 and some other markers associated with xanthophores, we showed that differentiated xanthophores are not ablated by the melanocyte-ablating drug neocuproine and this mitfa^+^aox5^hi^ subpopulation does not make new pigmented xanthophores following neocuproine treatment. However, current observations cannot definitively determine the potency and fates adopted by these cells. One possibility is that these cells are indeed progenitors that arise through cell divisions, are in an as yet undefined way lineally related to MP-0 and MP-1 subpopulations, and ultimately give rise to new melanocytes during additional rounds of regeneration. Given their expression of markers associated with multipotent pigment cell progenitors, these cells could be multipotent but fated toward the melanocyte lineage following melanocyte-specific ablation. However, we cannot exclude the possibility that these cells are another cell type. For example, there is a type of partially differentiated xanthophores that populate adult melanocyte stripes (McMenamin, et al., 2014). At least some of these cells arise from embryonic xanthophores that transitioned through a cryptic and proliferative state (McMenamin, et al., 2014). That the descendants remain partially differentiated could indicate that they are in more of a xanthoblast state and maintain proliferative capacity (Eom, et al., 2015). It is possible that some or all of the cells in question are melanocyte stripe-resident, partially-differentiated xanthophores that arise: (a) from cell divisions that are triggered by loss of interactions with melanocytes or, (b) simply to fill space that is vacated due to melanocyte death. Such causes for partially-differentiated xanthophore divisions have not been documented, but nonetheless this possibility must be considered given the mitfa and aox5 expression and proliferative potential of these cells. Transcriptional profiles of ‘cryptic’ xanthophores are not available to help clarify the nature of these cells. Lastly, the relationship between adult progenitor populations – MP-0, MP-1 and, potentially, mitfa^+^aox5^hi^ G2/M adj – and other progenitors present at earlier developmental stages is unclear and could be defined through additional long-term lineage tracing studies. In particular, previous examinations of pigment cell progenitors in developing zebrafish have identified dorsal root ganglion-associated pigment cell progenitors in larvae that contribute to adult pigmentation patterns (Singh, et al., 2016; Dooley, et al., 2013; Budi, et al., 2011). It is possible that these cells give rise to the adult progenitors we have identified. The further alignment of cell types that have been observed in vivo and cell subpopulations defined through expression profiling is a necessary route for understanding the complex relationship between stem and progenitor cells in development, homeostasis, and regeneration.”

Reviewer #2 (Recommendations for the authors):1) Identity/Xanthophores: The authors list 14 clusters that identify as xanthophores and call them mitfa+aox5+ cells on the UMAP (Figure 1). Later the authors follow mitfa+aox5+ cells dividing by scRNA-seq, and in the stripes and interpret these as McSCs (Figure 6, Figure 7). It seems likely these are simply dividing xanthophores? More evidence is required to link these cells to melanocytes, and melanocyte stem cells.

We agree with the editor and reviewers that the identities of the mitfa^+^aox5^hi^ cells and the interplay between these cells and the mitfa^+^aox5^lo^ cells is a fascinating, and originally unexpected, aspect of this manuscript. The issue, as we see it, is whether mitfa^+^aox5^hi^ cells that arise via cell division during regeneration are multipotent pigment cell progenitors or ‘cryptic’ xanthophores. The experiments we have performed to address this ambiguity have not worked for technical reasons, so we have tempered text in the relevant Results and Discussion sections to leave both options open. We have backed off from calling these cells progenitors but have included additional data showing that they (i.e. the mitfa^+^aox5^hi^ subpopulation of cells that we believe are daughters of mitfa+aox5hi cycling cells) express multiple markers associated with multipotent pigment cell progenitors that have been characterized in developing zebrafish. Our expanded Discussion is as follows below. Please also note that we have taken the reviewers advice and used the ‘progenitor’ label (MP = melanocyte progenitor) instead of ‘stem cell’.

“Heterogeneity may also be evident by the additional mitfa^+^aox5^hi^ G2/M adj subpopulation that likely arises via cell divisions during regeneration. There are reasons to think that this could be a progenitor subpopulation. Firstly, these cells arose in response to specific ablation of melanocytes. […] The further alignment of cell types that have been observed in vivo and cell subpopulations defined through expression profiling is a necessary route for understanding the complex relationship between stem and progenitor cells in development, homeostasis, and regeneration.”

2) scRNA-seq analysis: Please explain some of the choices taken in the scRNA-seq pipeline.a. Cells with 200 expressed genes are considered high quality. Can the authors justify this figure?b. We wonder whether the integration between WT and Kita datasets missed any other major differences between WT and mutant cells? Were the datasets so similar even before the integration?

A) Cutoff metrics are platform- and analysis software-specific. We adopted the cutoff of 200 expressed features as used by the developers of Seurat for analysis of CellRanger outputs from 10x platform 3’ prepared cells (https://satijalab.org/seurat/articles/pbmc3k_tutorial.html). A discussion of quality parameters that helped to guide our choices can be found in Ilicic et al. (https://www.ncbi.nlm.nih.gov/pmc/articles/PMC4758103/). We also filtered out cells expressing more than 10% mitochondrial reads and more than 50000 total reads. The current field standard is that cells with high mito reads in non-oncological tissues are likely dead or dying cells. Of note, we did try to relax this parameter during some initial analyses to identify dying melanocytes following neocuproine ablation; however, were unable to discern dying melanocytes from other dead or dying cells in the tissue preparation. We also found that cells with high numbers of unique genes expressed or high numbers of reads were likely to be doublets. This observation was helped by comparing our filtering strategy to the R package “DoubletFinder” (McGinnis et al., 2019). An example utilizing one of our WT Day 0 (Author response image 1) demonstrates the detection of high read and high gene number cells as doublets.

**Author response image 1. sa2fig1:** 

B) The reviewer brings up an interesting point about possible forced similarity between the zebrafish strains. We clustered the datasets separately and show the UMAPs in Figure 4—figure supplement 2. Of course, the UMAPs are not identical, and due to the reduction of certain cell subpopulations in kita(lf) mutants, namely direct differentiation and cycling differentiation, we do not expect them to look the same. There are also some differences in the aox5hi grouping of subpopulations. Nonetheless, there is striking conservation of cell types/clusters. Specifically, similar subpopulations were identified, and were, for the most part, positionally similar to each other in the UMAP. None of the differences observed, aside from the noted differences in direct and cycling differentiation subpopulations, are a focus of our manuscript.

3) Statistics: Although the authors used appropriate statistical tools in the analysis of the scRNA-seq datasets, we are concerned about the stats used in the plots generated with GraphPad Prism. The Student's t-test can be used when comparing the mean of two datasets with normal distribution. However, it cannot be used for multiple comparisons. Please review the choice of their statistical test for Figures 3C, 4A, 4E, 5B, 5D, 5F, and S5B, and indicate the missing statistical test in Figures S1C, S2D, S6B.[Please use ANOVA or similar for initial analyses to assess whether significant differences are present overall, and appropriate post hoc comparisons, like Tukey-Kramer, if warranted. Include values of test statistics (e.g., F) with degrees of freedom for overall tests. -Parichy]

We thank the reviewer for their careful eye, we have reviewed all the statistical tests presented in the paper and made the following changes:

F3C: We now utilize a one-way ANOVA comparing Day 0 to other time points.

F4B: We now utilize a one-way ANOVA comparing WT, kita(lf) and kitlga(lf).

F4E: We have re-graphed these data as a bar graph to make comparisons more evident. We now utilize a one-way ANOVA comparing WT, kita(lf) and kitlga(lf).

F5B: We utilize a Student’s t-test.

F5D: We utilize a Student’s t-test for each time point.

F5F: Here we still use a Student’s t-test since kita(wt) and kita(lf) samples are considered separately, each with its own control group

S5B: Here we still use a utilize a student’s t-test to compare two means in a pairwise fashion.

S1C: This figure is intended for qualitative visualization of the presence of different cell types at different time points. We make no comparisons between samples in this panel. More in depth description of population dynamics, with statistical support, is investigated in Figure 2.

S2D: This figure is intended to show qualitative similarity between two replicates and so we have not included a statistical test.

S6B: This panel lacked calculations of statistical significance. We have now added Wilcox Rank Sum test values.

4) Context in the field:a. The authors directly examine adult skin, which has not been done in depth before, and provide an important dataset resource. However, it is important to interpret the data within the context of the wider melanocyte stem cell field (i.e. DRG-associated and daughter cells lining the peripheral nerves as shown by Budi et al., 2011; Singh et al., 2014; Singh et al., 2016, Brombin et al., 2022). These cells are also kit-dependent (Dooley et al., 2013). It is important to address how the melanocyte adult skin progenitors relate to the DRG-associated progenitors. One idea is that the cells that line nerves are a source of melanocytes. Not all the pigment progenitors described by Saunders and colleagues are skin-associated. Could these progenitors contribute to the regeneration in adult fish as previously shown (Budi et al., 2011)?

The ability of DRG- or nerve-associated melanocyte stem or progenitor cells to contribute to adult regeneration is an interesting possibility and something we have considered:

a) Firstly, the DRG- and nerve-associated progenitors that have been recognized at earlier stages of development (embryonic and juvenile stages) have not been tracked to determine lineal relationships to the adult progenitors we have characterized. Lineage-tracing experiments could potentially make a link between earlier melanocyte progenitors and the cells we find in adult skin.

b) If nerve-associated progenitors or progenitors in other deeper-lying locations were present in adults (i.e. a separate population from what we have defined through scRNAseq) could they contribute to melanocyte regeneration? In our serial imaging, we found that all new melanocytes arose from skin-resident progenitors. Thus, new melanocytes from regeneration come from skin-resident progenitors. What is less clear is whether new skin-resident progenitors can be derived from migration into the skin of deeper-lying cells. This could potentially occur rapidly or slowly after a new complement of melanocytes have been regenerated and would be evident if any cells that migrated after one round of regeneration produced melanocytes in a subsequent round of regeneration. We have not conducted serial imaging over such an extended period of time to address this question.

The experiments needed to relate DRG- or nerve-associated progenitors to skin-resident adult progenitors are challenging, if at all possible. However, this is an interesting question, and we have included the following in our discussion to add context as the reviewer has recommended:

“Lastly, the relationship between adult progenitor populations – MP-0, MP-1 and, potentially, symmetrically dividing cells – and other progenitors present at earlier developmental stages is unclear and could be defined through additional long-term lineage tracing studies. In particular, previous examinations of pigment cell progenitors in developing zebrafish have identified dorsal root ganglion-associated pigment cell progenitors in larvae that contribute to adult pigmentation pattern (Singh, et al., 2016; Dooley, et al., 2013; Budi, et al., 2011). It is possible that these cells give rise to MP-0, MP-1 and/or symmetrically dividing subpopulations. The further alignment of cell types that have been observed in vivo and cell subpopulations defined through expression profiling is a necessary route for understanding the complex relationship between stem cells in development, homeostasis, and regeneration.”

b. We have concerns with the use of the term "stem cell", and believe the data is more in line with "Progenitor" as they have previously called these cells in Iyengar et al., 2015.

In retrospect we are comfortable changing to the “progenitor” label. The conventional definition of a stem cell includes both the ability to create differentiated descendants and self-renew. Our data indicate some cells have the capacity to produce new melanocytes, but the ability of other cells to self-renew has not been formally demonstrated by our data. In the revised manuscript we have renamed McSC-0 and McSC-1 subpopulations MP-0 and MP-1, respectively.

c. For their cycling differentiation cells, it might be appropriate to cite other work showing division of differentiating melanocytes as well (e.g. Taylor et al. 2011)

We are unsure how to relate the cycling differentiation cells to those defined by Taylor et al. While the Taylor study nicely shows divisions of pigmented, differentiating melanocytes during melanocyte regeneration in embryos, we don’t find such cells in adults. Specifically, cells that divide during melanocyte regeneration in adult skin are never pigmented. Furthermore, the cell divisions in adults that produce new melanocytes are asymmetric, in which one daughter differentiates and the other remains undifferentiated. Because of these fundamental differences, we feel that a comparison between pigmented, dividing melanocytes in embryos and cycling differentiation cells in adults is too uncertain and have not included it.

5) Melanoma: As the authors rightfully claim, the study of McSC might be beneficial to the understanding of how these contribute to melanoma, however, they do not make an actual link to melanoma signatures. Perhaps they could cite and discuss some of the papers that show McSCs and progenitors in zebrafish are relevant to melanoma?

We thank the reviewer for this suggestion. We agree the overlap between MP signatures and melanoma is a fascinating facet of studying MPs. To demonstrate the link between MPs and melanoma we have bolstered our discussion of the potential utility of examining the overlap between MPs signatures during regeneration and melanoma as follows:

“In melanoma, stem cell-like identities are implicated in tumor maintenance and drug resistance. In single-cell analysis of human melanomas, a subpopulation of cells expressing high levels of AXL, a putative cancer stem cell marker, and low levels of MITF were present in treatment-naïve human melanoma samples (Tirosh, et al., 2016). These AXL-high MITF-low cells were admixed with AXL-low MITF-high differentiated cells, demonstrating cellular heterogeneity present in untreated melanomas. Furthermore, samples from patients after relapse were enriched for this AXL-high state, furthering the concept that stem-like cells facilitate relapse. Similarly, zebrafish melanoma models have demonstrated the role of a stem-like minimal residual disease cells in facilitating relapse (Travnickova, et al., 2019). In this study, MITF-low zebrafish melanoma cells had striking transcriptional similarities to invasive melanoma signatures found in humans following treatment with small molecule inhibitors of MEK or BRAF (Travnickova, et al., 2019). Additional studies have found stem-like properties in melanoma cells involved in therapeutic resistance and tumor relapse (Boshuizen, et al., 2020; Mehta, et al., 2018; Rambow, et al., 2018; Sharma, et al., 2017; Konieczkowski, et al., 2014). Our study provides a reference to which melanoma cells, as well as cells involved in pigmentary diseases such as vitiligo, can be compared to for identifying relationships between and conserved pathways shared by melanocyte progenitors and cells involved in melanocytic diseases.”

We have also compared signatures of MP and other subpopulations to signatures defined for melanomas from humans (Tirosh et al.) and zebrafish (Brombin et al., Travicknova et al.). These comparisons were conducted in two ways: (a) MP and other subpopulations compared to individual tumors, and (b) MP and other subpopulations compared to subpopulations of tumor cells defined after pooling all tumor cells in a given study. From the first comparisons, it is clear that there is marked heterogeneity between tumors. Some tumors have signatures that are aligned with MPs, whereas other tumors have signatures more similar to melanocytes, cycling differentiation and direct differentiation subpopulations. From the second comparisons, there are subpopulations of tumor cells (derived from multiple different tumors) that are aligned with MPs and other subpopulations of tumor cells with signatures more similar to melanocytes, cycling differentiation and direct differentiation subpopulations. From this latter type of comparison, we suspect that the tumor subpopulations more similar to MPs are likely to be tumor stem or propagating cells. Further functional would need to be performed to solidify this hypothesis.